# Navigating Pyrolysis Implementation—A Tutorial Review on Consideration Factors and Thermochemical Operating Methods for Biomass Conversion

**DOI:** 10.3390/ma17030725

**Published:** 2024-02-02

**Authors:** Waheed A. Rasaq, Charles Odilichukwu R. Okpala, Chinenye Adaobi Igwegbe, Andrzej Białowiec

**Affiliations:** 1Department of Applied Bioeconomy, Wrocław University of Environmental and Life Sciences, 37a Chełmońskiego Str., 51-630 Wrocław, Poland; waheed.rasaq@upwr.edu.pl (W.A.R.); ca.igwegbe@unizik.edu.ng (C.A.I.); 2UGA Cooperative Extension, College of Agricultural and Environmental Sciences, University of Georgia, Athens, GA 30602, USA; charlesokpala@gmail.com; 3Department of Chemical Engineering, Nnamdi Azikiwe University, Awka 420218, Nigeria

**Keywords:** thermal conversion process, feedstock, temperature, pressure, process reactors, learners

## Abstract

Pyrolysis and related thermal conversion processes have shown increased research momentum in recent decades. Understanding the underlying thermal conversion process principles alongside the associated/exhibited operational challenges that are specific to biomass types is crucial for beginners in this research area. From an extensive literature search, the authors are convinced that a tutorial review that guides beginners particularly towards pyrolysis implementation, from different biomasses to the thermal conversion process and conditions, is scarce. An effective understanding of pre-to-main pyrolysis stages, alongside corresponding standard methodologies, would help beginners discuss anticipated results. To support the existing information, therefore, this review sought to seek how to navigate pyrolysis implementation, specifically considering factors and thermochemical operating methods for biomass conversion, drawing the ideas from: (a) the evolving nature of the thermal conversion process; (b) the potential inter-relatedness between individual components affecting pyrolysis-based research; (c) pre- to post-pyrolysis’ engagement strategies; (d) potential feedstock employed in the thermal conversion processes; (e) the major pre-treatment strategies applied to feedstocks; (f) system performance considerations between pyrolysis reactors; and (g) differentiating between the reactor and operation parameters involved in the thermal conversion processes. Moreover, pre-pyrolysis activity tackles biomass selection/analytical measurements, whereas the main pyrolysis activity tackles treatment methods, reactor types, operating processes, and the eventual product output. Other areas that need beginners’ attention include high-pressure process reactor design strategies and material types that have a greater potential for biomass.

## 1. Introduction

The environment today is confronted by an unending cascade of global anthropogenic and ecosystem-based challenges. Biomass is considered to have the potential to be utilized as an alternative energy source. The conversion of the high carbon content of biomass through thermochemical treatment resulted in better fuel properties of biochar production. Pyrolysis is the most researched thermochemical technique in the past decade among the few well-established methods for treating biomass and biogenic waste in order to produce high-quality and yield energy products such as biochar, bio-oil, and pyrolytic gas. The obvious aftermath of the industrial revolution brought about a steady geometric increase in population growth, which noticeably altered the balance of global carbon. The global population in 2013 was estimated at 7.2 billion and is estimated to increase by a billion in 2025; this makes the energy demand that is required for agricultural, industrial, and transportation development very crucial [1]. Global waste management, on the one hand, is among the United Nations’ Millennium Development Goals (MDGs) from 2000, and particularly key here is its 7th Goal: To Ensure Environmental Sustainability, which subsequently progressed in the 2008 Waste Framework Directive [2], which aims to improve (waste) management strategies in accordance with the Sustainable Development Goals (SDG) [3]. On the other hand, waste disposal methods like, for example, composting should be effective in helping to reduce greenhouse gas emissions [4]. Moreover, the generated bio-waste materials would noticeably vary, especially in composition, largely due to some factors like the type of community and its consumers, industrialization, institutions, and commercial entities [2]. A great portion of bio-waste material, and how it transforms into biofuel, as well as other energy sources remains a major research focus, especially from the environmental standpoint [5]. Biomass, however, is increasingly considered to be the potential alternative renewable energy source. Pyrolysis, among the initial stage(s) in gasification, would help utilize biomass energy, alongside other thermochemical conversion procedures. In addition to the prevailing environmental issues, there are other pressing biomass-related challenges that involve many pyrolysis-based studies. Besides that, the context of thermal conversion technology (i.e., determination of operating parameters of pyrolysis-based) and reactor types have been based on the desired characteristics of the product (bio-oil, biochar, and pyrolytic gas), as well as on the field of biomass pyrolysis and upgrading. Feedstock properties, product characteristics, reactor type, and upgrading options are among the key areas demonstrated in the synthesized literature, providing relevant information. It is worth mentioning that some conducted reviews have looked at the systematic approaches for mapping biomass resources to conversion pathways, forming the basis for biomass valuation and informing when biomass pre-processing is needed in order to ensure feedstocks are ready for conversion [5]. Furthermore, bio-oil derived from pyrolysis biowaste would serve as chemicals/fuel products. The production and composition of pyrolysis oil are affected by the biomass composition and process operating parameters [6].

In recent decades, the research momentum about pyrolysis and related thermal conversion processes is on the rise, involving a wide range of biomass/feedstock targets. For emphasis, pyrolysis simply depicts the use of heat treatment to bring about an irreversible chemical change in the absence of oxygen, specifically [5,7,8]. Moreover, pyrolysis remains one of the most efficient techniques for thermochemical conversion without the involvement of oxygen. This process yields carbon-enriched hydrocarbons (bio-oils), biochar, and volatile gases containing molecules that are rich in oxygen and hydrogen [9]. Generally, the major pyrolysis products include biochar, bio-oil, pyrolytic gas, and tar, among others, which largely depends on the process type, whether it is slow, fast, or flash, considering their tightly linked technological/product yield components [10,11,12,13,14,15,16]. The harnessing of the associated biomass energy via thermochemical processes should be eco-friendly and should be completed with solid waste conversion technologies at high temperatures [17]. If the target product(s) is to be achieved, a thorough prior knowledge and understanding of different biomasses, as well as the conditions/situations of their pyrolysis is warranted [5]. For the temperature of pyrolysis to achieve a target level, the specific energy demand largely depends on biomass moisture and the process temperature and duration [18]. To reiterate, the pyrolysis type would directly connect with the reactor types [19,20,21,22,23]. To learn the pyrolysis operating process, the description of reactors alongside their peculiarities that make them well suited for one or more biomass feedstocks is also warranted, as this has a direct influence on the type of the anticipated product output.

In addition to the heating methods and their reactor types, biomass properties can affect the pyrolysis process [24,25,26], especially when considering the wide range of decomposition processes and realized products (of biomass) [27]. In addition to the different biomasses and their prevailing conditions, understanding the intentions of specialists/stakeholders to engage in pyrolysis/thermal conversion activities emanates from the quest to achieve a desirable end-product (of a given biomass/feedstock). Some major advantages of pyrolysis can include the following: (a) a high degree of efficiency and profitability, as well as the suitability to convert a wide range of solid waste into storage energy; (b) the minimal nature of greenhouse gases like HCl, NO_x_, and SO_x_; (c) the absence of corrupt organic matter in the pyrolyzed residue to prevent the extraction of metal substances via solvent; (d)pyrolysis is capable of processing garbage waste that is not suitable for landfill and incineration; and (e) the fixing of harmful components like heavy metals and sulfur that are present in the (raw material) waste. Some disadvantages, however, can include: (a) the waste processing, if not properly developed, could still pose environmental problems, and (b) to implement the process in a large-scale pyrolysis project will require a permit by the government, given the differences in the prevailing policies [28,29,30]. Despite the abovementioned advantages and disadvantages associated with thermal conversion, there still remains some research concerns. For examples, what are the primary essentials that a thermal conversion enthusiast, especially a beginner, needs to grasp/understand regarding the implementation of a typical pyrolysis-based study? If this question were to be answered, another fundamental question could be guided by the following: (a) Why is it important to shed more light on implementing a typical pyrolysis-based study? (b) Could it be to strengthen the subject area? (c) Could it be to attract more enthusiasts into becoming more engaged in pyrolysis-based studies? (d) Additionally, if the implementation process were to be better understood, what would be the benefit(s)? (e) Could it enable/help new investigators increase their proactivity, as well as emerge better engaged in any given pyrolysis-based study? (f) Could it enable/help in enhancing their creativity, as well as their initiative of research ideas/questions for the implementation activity of pyrolysis-based studies? These above-mentioned questions underscore the justification/rationale of why the authors herein have deemed it needful to conduct a captivating review synthesis in order to support the existing information and to seek how to navigate pyrolysis implementation, specifically when considering the factors and thermochemical operating methods for biomass conversion, drawing the ideas from: (a) the evolving nature of the thermal conversion process; (b) the potential inter-relatedness between individual components affecting pyrolysis-based research; (c) pre- to post-pyrolysis’ engagement strategies; (d) the potential feedstock employed in the thermal conversion processes; (e) the major pre-treatment strategies applied to feedstocks; (f) the system performance considerations between pyrolysis reactors; and (g) differentiating between reactor and operation parameters involved in the thermal conversion processes. In addition to pre-to-main pyrolysis stages and their respective analyses, either developed or adopted from other fields, this tutorial review has provided some understanding about the operational standpoints, including (a) that pre-pyrolysis activity involves biomass selection and analytical measurements, and (b) that the main pyrolysis activity involves treatment methods, reactor-types, operating processes, and eventual product outputs, as shown in Figure 1.

## 2. The Evolving Nature of Thermal Conversion Process

Previous findings involving the implementation of thermal conversion based-studies on conditions/situations and different biomasses, along with how these experimental aim(s)/objective(s) of the various studies were developed and the respective subsections captured are all present in Table 1. Additionally, the pyrolysis-based studies shown engaged with varying aims/objectives. More so, to carry out the experimental procedures developed on the basis of pyrolysis, strong considerations need to be given to the parameters involved, such as the materials, the characteristics of biomass samples, the sample preparation and pyrolysis, the economic analysis, as well as validation via the experimental procedures. Additionally, there could be various thermo-kinetics of the feedstock, which would associate with the thermal operating conditions, if high quality pyrolytic products like biochar, bio-oil, and pyrolytic gas were to be achieved. Tian and colleagues’ experiments that were conducted on rice husks were carried out using two pyrolysis-coupled real-time volatile monitoring techniques (TGA–FTIR and Py-GC/TOF-MS). The findings demonstrated that in the temperature range of 200 to 330 °C, 330 to 390 °C, and 390 to 600 °C, respectively, rice husks showed three mass loss and gaseous product evolution stages. It was shown that 2,3-dihydro-benzofuran was the main hemicellulose product after speculating on the formation pathways of the 24 main volatile species. On the other hand, 2-methoxy-4-(1-propenyl)-phenol was a potentially key active intermediate and was highly unstable during the pyrolysis of the lignin constituent in RHs [31]. Besides the acquisition of knowledge regarding primary volatile compositions, the behavior of mass loss in a given feedstock, as well as the reaction kinetics properties [31,32] are important. For instance, seven partners participated in an international round-robin study, conducting TGA pyrolysis experiments on pure cellulose and beechwood at various heating rates. The activation energies of cellulose, hemicellulose, and conversions of up to 0.9 with beechwood showed deviations of about 20–30 kJ/mol in all experiments [33], feedstock preparation via the adoption of the biochar catalyst method upgrading options, hybrid pretreatment methods, and the comparisons of untreated and hydrochloric acid treatment of various biomass feedstock [34,35,36]; these are all examples of studies where feedstock was directly associated with the thermal operating conditions. In Téllez and colleagues’ study, using lab-scale fast pyrolysis in a vacuum, rice husks (RHs) were converted into pyrolytic oils, enriched with levoglucosan (LG). They investigated how the pretreatment of the biomass and the pyrolysis temperature (300–700 °C) affected the yields of pyrolysis products and the selectivity for the LG formation. RHs pretreated with hydrochloric acid at 400 °C produced a maximum oil yield of 47 wt.%, which was 1.4 times more than the amount of oil produced at the same temperature from untreated RHs [36]. Also, activated carbon would help purify the bio-oil organic compounds, which could lead to environmental pollution [37]. Besides the thermal conversion operating parameters, like temperature [38] alongside the catalyst sorbent addition, there is the application of the Coats–Redfern method that could impact the end products’ properties. Thus, understanding the influence of temperature on the evolution of the structures and the organic content of biochar [39,40,41,42,43] is key. The co-pyrolysis of rice straw (RS) and *Ulva prolifera* macroalgae (UPM) was investigated by Hoa et al., using a range of activated biochar catalysts supported by nickel-iron layered double oxides (NiFe-LDO). The bio-oil yield from co-pyrolysis was higher than that from individual pyrolysis. At 500 °C, the biomass mixture of RS/A-UPM produced the highest bio-oil yield (46.68 wt.%). However, the combination of RS and UPM without acid-treated UPM demonstrated a reduced bio-oil yield. Because of the coke formation during the catalytic pyrolysis up-gradation, the bio-oil was reduced. However, using the 5% Ga/NiFe-LDO/AC catalyst improved the bio-oil quality [39]. The correlations of pyrolysis characteristics with biomass types should be considered alongside the associated mechanisms [44]. A bio-fuel could be upgraded by various thermal conversion methods from the feedstock [45,46]. Furthermore, conventional thermogravimetric analysis could be applied to investigate the mechanism interaction of the co-pyrolysis process [47], which might offer fresh perspectives for eco-innovative circular economy solutions [48].

## 3. Potential Inter-Relatedness between Individual Components Affecting Pyrolysis-Based Research

The application of the thermal conversion process requires considering numerous factors, including the specific reactors, the corresponding biomass/feedstock, the choice of research/objective, the target products, the feedstock type, the funding for research, the time/period for study, and the geographical location. The potential inter-relatedness of the individual components affecting pyrolysis-based research, presented in Figure 2, requires some considerations prior to the design of an experiment/methodology. Several aspects that require focus include the research objective, process cost based on the plant size, the types of reactors, the extents of supply and feedstocks, and the experiment location. Other pyrolysis-related components that directly relate to the pyrolysis process operational costs would depend on the size, quality, reactor types, and the laboratory/enterprise, wherein the experiment is performed. However, technology is just one aspect of innovation for more waste-based sustainable thermal systems, which should provide systematic yet innovative solutions towards a more resource-efficient economy with waste management [48].

A number of pyrolysis-based investigations have had to consider the cost and time required for the pyrolysis process, according to Ringer et al. [85], which provided a broad perspective of pyrolysis technology in converting biomass material to bio-oil, and other valuable products. It was presented through a thorough technical and financial analysis of a plant that could produce 16 tons of bio-oil per day [85]. The Circulating Fluidizing Bed (CFB) reactor type is able to provide a high-quality product yield, and a solid foundation for scaling up and for a high-quality product yield was identified, which estimated the investment and operating costs for 550 tons per day of moisture ash free (MAF) biomass, with a 48.3 MM capital investment and an estimated total installation cost of 28.4 MM USD [85]. Suntivarakorn et al. used a Circulating Fluidized Bed Reactor (CFB) with sand as the bed material in order to study the production costs of pyrolysis oil production from Napier Grass. The maximum oil production from pyrolysis was 36.93 wt. percent, demonstrated at bed temperatures of 480 °C, superficial velocities of 7 m/s, and feed rates of 60 kg/h. Based on pyrolysis oil production properties, water content, density, heating value, viscosity, and pH, the results were 48.15 wt.%, 1274 kg/m 19.79 MJ/kg, 2.32 cSt, 2.3, respectively. Additionally, the values of total energy conversion and cold efficiency to pyrolysis oil were 19.77% and 24.88%, respectively. The energy used in the heating process was the source of the bulk of energy consumption, with an estimated pyrolysis oil production cost of 0.481 $/L at 75 kg/h of feed rate [86].

Furthermore, the pyrolysis-based study as a choice of (research) objective would be crucial, which potentially connects the individual components affecting the (pyrolysis-based) research and the other related factors associated with the procedure, reactor type, feedstock types, the time required, target products, and the application method of the experiment. If the experiment undergoes fast pyrolysis, in which less time is required when compared to the low-temperature thermal process type, selecting the process type would depend on the desired products, especially where all the three possible common products (bio-oil, biochar, and pyrolytic gas) are targeted. For instance, Wu et al. performed an experiment with three different feedstocks, namely rice, maize, and wheat straw, with the same pyrolysis reactor (rotating bed) operating at five different temperatures from 300–700 °C, where the rice, maize, and wheat straw product yield were observed at 500 °C, accordingly: (a) liquid: 37.02, 38.91, and 35.89%; (b) char: 38.25, 34.04, and 35.25%, and (c) gas: 26.73, 27.94, and 28.86, respectively [87]. Generally, the thermal conversion process requires more time in each step before pyrolysis products can be achieved.

## 4. Pre- to Post-Pyrolysis’ Engagement Strategies

The knowledge and understanding needed for pyrolysis connects largely with reduced gas emissions and its implementation cost together with its small-scale nature. Prior to biomass selection and its components, the availability and location should be considered. Before identifying which reactor type to use, the biomass feedstock materials, the required specificities associated with a reactor, and the visualization of the anticipated product, together with the (reactor) energy demand should all be considered. Prior to anticipating the target pyrolysis-based product, consideration should be given to the operating conditions, the intricacies associated with the thermal conversion process, and any (internal/external) influencing features (Refer to Figure 1).

Particularly in the context of pyrolysis implementation, the constituents of some selected recent experimental works revealing the pre-to-main pyrolysis stages, respectively, from biomass selection and analytical methods, to the biomass treatment methods, reactor types, operating process, and product outputs are shown in Table 2. Besides the differing specific objectives, differences would still emerge in the contents of both pre- and main pyrolysis stages. However, there could arise some situations where the feedstock and its pretreatment reflect each other. Furthermore, the individual study shows specific objectives, which in turn would either directly or indirectly determine the subsequent experimental method/design requirements. For example, corn stalks were among the biomass selected for use by two workers [88]; despite this, both studies clearly had different objectives, hence different study design approaches. On the one hand, the study of Zhu and colleagues determined the recovery efficiency of minimizing VFAs and sugars at different HTS and cornstalk structure characterizations, which employed a batch reactor and an HTC operating process, wherein the product output included sugars and volatile fatty acids (VFAs). According to their findings, 92.39% of aqueous products had the highest recovery of reducing sugars and VFAs, which is equivalent to 34.79%, based on dry biomass. In addition, significant changes in organic groups at different HTS were identified through FTIR and TGA, and, as HTS parameters increased, the cornstalk’s structure gradually changed from stiff, highly ordered fibrils to a molten and grainy structure, via SEM [88]. On the other hand, the study of Wang et al. explored the corn stalk performance of the wet torrefied sample’s performance in biomass pyrolysis polygeneration, wherein the fixed-bed reactor was employed, and which had biochar as the output product. Aside from the above-mentioned, all the studies shown in Table 2 appear to have resembling analytical methods, which largely involved moisture, organic matter, and ash content, with very few exceptions [15,25,52,54,60,65,83,88,89,90,91,92,93,94,95,96,97,98,99,100,101,102,103]. Palamanit and colleagues used an agitated bed pyrolysis reactor to examine the yields and characteristics of pyrolysis products obtained from oil palm fronds, trunks, and shells. The pyrolysis temperatures of 400, 450, and 500 °C were applied to these feedstocks. The findings demonstrated that the pyrolysis temperatures and varieties of oil palm biomass had an impact on the yields and characteristics of the final product. The maximum liquid yield was obtained from oil palm fronds pyrolyzed at 500 °C. The HHV of the liquid and biochar product was 18.95–22.52 MJ/kg and 25.14–28.45 MJ/kg, respectively. Furthermore, the SEM result demonstrated that the produced biochar had a porous structure surface with a surface area of 1.15–4.43 m^2^/g [103]. Moreover, the biomass treatment method involved chemical [25,102,104], hybrid [35], and physical and thermal [15,52,88,97,98,99,100,101] types. For instance, TGA–FTIR and PY-GC/MS were used to investigate the reaction mechanism for the pyrolysis of cellulose, hemicellulose, and lignin in the presence of CaO. The results showed that CaO would react with acids and phenols from hemicellulose pyrolysis, sugars from cellulose pyrolysis, and phenols from lignin pyrolysis at low temperatures (400–600 °C). However, at higher temperatures (600–800 °C), the CaO catalytic effect was more noticeable. Specifically, CaO facilitated the catalytic decarbonylation of ketones to form CO during hemicellulose pyrolysis, while also increasing the formation of hydrocarbons. Additionally, CaO addition promoted radical reactions during lignin pyrolysis, increasing the CH4 yield [25]. Duman and Janik attempted to enhance the production of hydrogen from the steam pyrolysis of olive pomace in a two-stage fixed-bed reactor system, where various char-based catalysts were evaluated. The catalysts included biomass char, nickel-loaded biomass char, nickel or iron-loaded coal chars, and coal char used as catalysts. Thus, BET, XRD, XRF, and TGA were used to characterize catalysts. Their results showed that the steam obtained without a catalyst had no influence on hydrogen production, and the production of hydrogen was improved when the temperature increased from 500 °C to 700 °C, when both Ni-impregnated and non-impregnated biomass char were present [100]. Additionally, the differences in the study objectives of the various researches produced various output products, like biochar, bio-oil, pyrolytic gas [15,54,101,102], sugars and volatile fatty acids (VFAs) [88], hydrogen [100], char, phenols, and anhydro sugars [94], glucose [104], and furan [96]. The reactor selection and operating conditions appear to connect with the feedstock and its resultant product output, as well as the preparatory materials required before or during the engagement of pyrolysis/thermal conversion-based study.

## 5. Potential Feedstock Employed in Thermal Conversion Processes

### 5.1. Feedstock Composition by Various Thermo-Chemical Reactors

A summary of different biomasses/feedstocks as classified by various thermo-chemical reactors (considerations/factors of research objective) is presented in Table 3. The various biomass/feedstock types and their commercial importance within the energy sector aspects of sustainability/sustainable development goals are vital. In the thermal process, this can be divided into a number of different types, such as lignocellulosic biomass, municipal solid waste, and fuel derived from refuse, and how its properties affect the pyrolysis process parameters. Due to its potential to serve as a bio-renewable source of fine/commodity chemicals and fuels, we focused on all potential feedstocks for thermal conversion activities [11] with both a single and a combination of reactors. Feedstock pretreatment is also important and is required in many cases to achieve the high quality and quantity of pyrolytic target product, and the treatment could be performed through different methods, such as chemical pretreatment, physical pretreatment, thermal pretreatment, biological pretreatment, and hybrid pretreatment with suitable reactors to achieve the best quality end product in any thermal conversion study (Refer to Table 2). But, in order to have a clear understanding of how the main biomass decomposes, it is important to be aware of its features and structure, especially in relation to the moisture content and the precise temperatures needed to produce various pyrolytic products. Furthermore, the pyrolysis process’ temperature variations reflect various layers, including the hemicellulose, cellulose, and lignin of biomass structure [27]. Given that it is known that these three main layers pyrolyze at various temperatures range (200–300; 300–350; and 350–500 °C), respectively, the emergent products, such as chemicals, fuels, and materials through different biochemical and thermo-chemical processes, would be achieved at specific temperature points [27,106]. Previous biomass samples were investigated by Bahcivanji et al. [67] and the pyrolysis yield of hydrochar at 350 °C for 5 h was comparable to the pyrolysis yield of waste biomass using the same experimental conditions, when compared to the direct pyrolysis of waste biomass via the HTC process. Only when the pyrolysis temperature was raised to 550 °C for 5 h did the pyrolysis yield of the feedstock fall below that of hydrochar. The higher the temperature of pyrolysis (from 350 to 550 °C) and the duration time (from 1 to 5 h), the more microporosity was produced, while the phytotoxicity was decreased [67]. In addition, similar results were obtained by numerous studies, where the hydrochar showed a lower pH than the original feedstock [80,107].

### 5.2. Importance of Feedstock Composition in the Thermal Conversion Process

The use of biomass wastes as a fuel source has drawn significant attention in the green society and in environmental management. Therefore, the typical composition of the feedstock group in the thermal conversion process is shown in Figure 3; the groups of feedstocks subjected to the thermal conversion process and the possibility of the combination/mixture of them to obtain a high-quality yield of the target products is described in Figure 3. From the available feedstock, 10 clusters were identified, with woody biomass being the dominant, although we were more focused on biomass materials in this study. However, it clearly shows in the map that the group of woody feedstock is more likely to appear with a high percentage of oak, followed by other feedstock types, including shell-nut, corn stalk, rice straw, coffee husk, banana leaves, poultry manure, garden material, and fruit. More so, several related studies investigated such feedstock(s) for different end products targets, and this is necessary to shed more light on this. It is very possible to perform any kind of thermal conversion process in combination with different feedstocks while considering the thermal and kinetic characterization for the target end product; this can be performed by knowing the physical and chemical properties of the material, thus allowing for the right selection, based on the characteristic properties of the individual types.

Furthermore, Braz and Ribeiro [108] investigated a mixture of sewage sludge with pruning residues in a proportion of (50 mass%); they compared the results with the sewage sludge without a mixture to determine the thermal and kinetic characterization of the samples. The result shows that the average activation energy value of the sewage sludge sample and of the mixture, respectively, was 219 and 161 kJ mol^−1^, supporting the incorporation of pruning residues in the sewage sludge. In the degradation process, a remarkable increase in activation energy was observed, which ranged from 20–70% via conversion in the sewage sludge sample, despite the almost linear behavior noticed within the mixture decomposition reaction [108]. For some time, the process of combining biomass with other wastes for a power generation purpose has been studied as a way of reducing the waste material disposed into landfills, which involve the mixture of biomasses, such as pine, eucalyptus, sawdust, chestnut, pulp waste, grape, and coffee husks, all of which have aimed to choose the best raw materials for making pellets that were available in the study area. Furthermore, blends of pine sawdust with 10–30 percent chestnut sawdust were considered best for pellet production [109]. Elsewhere, Lajili and colleagues, to measure the moisture, ash content, bulk density, and heating values, made agropellets from olive waste, a by-product of an olive mill, which was mixed in various ratios with sawdust from pine trees. Olive waste’s high moisture content decreased during the process, and each chosen sample’s ash content was found to be in compliance with the recognized French agropellets standards [110].

Additionally, Boumanchar et al. conducted a study where parameters were evaluated for various abundant materials (including two types of biochar, different biomasses, synthetic rubber, cardboard as a potential municipal solid waste, and plastic). The objective was to contrast the calorific value of each substance when used separately with the combined experimental and theoretical HHVs of the two substances. Various mixtures in proportions of 25/75, 50/50, and 75/25 percent were prepared. The experiment’s findings revealed that the heating values of lignocellulosic materials ranged from 12 to 20 MJ/kg: 13 MJ/kg for cardboard, 27 and 32 MJ/kg for the first and second batches of biochar, respectively, and 37 and 38 MJ/kg for plastic and synthetic rubber [111]. Furthermore, the biomass mixture of feedstock was studied using a reaction vessel for the HTC process. HTC was performed on rice hulls, Loblolly pine, Tahoe mix (Jeffrey pine and white fir), corn stover, and switch grass. The results showed that the energy densification of biomass increased up to 43%, and the reaction temperature significantly impacted the energy densification and mass yield. The production of hydrochar increased the fixed carbon and decreased volatiles at a process temperature of 260 °C [112].

## 6. Major Pre-Treatment Strategies Applied to Feedstocks

Feedstock pretreatment is very important during any form of thermal process to remove or change the biomass components and to improve the target product’s quality. And this can be completed in different ways, such as physical, chemical, thermal, and hybrid treatment [113], and the description of each method is as follows:(a)Physical pretreatment

The first step of biomass feedstock pretreatment as a preliminary to feeding into a pyrolysis reactor is grinding the particle according to the reactor requirement particle size for the perfect process. Since the biomass thermal conductivity is very low (about 0.1 W/(mK), then the biomass pyrolysis mechanism might be affected by the temperature gradient across the particles in the process. Therefore, quick heating to achieve the target pyrolysis temperature level is difficult, and the only way to accomplish the target is to reduce the particle size to much smaller sizes. Usually, biomass particle size depends on the reactor type in the pyrolysis process; for instance, a fluidized bed requires 2–5 mm in biomass particle size, and some reactors require much larger particle sizes. Importantly, if the biomass particle size is bigger than the reactor requirement, it could result in less bio-oil and a higher char production yield, respectively, because biomass might partially be pyrolyzed [113].

(b)Chemical pretreatment

This is one of the pretreatment processes of biomass involving the use of liquid solvents for washing or cleaning, such as an acid or water, for the purpose of eliminating minerals or inorganics materials in biomass. According to Blasi et al., straw’s pyrolysis properties are affected by washing with water; this increases the bio-oil yield, while the char production decreases [114]. A similar experiment investigated by Carrier et al. [115] showed the application of an acid as a biomass washing, such as HNO_3_ and HF, resulted in a reduction of the mineral content of biomass [115].

(c)Thermal pretreatment

The thermal pretreatment of biomass is achieved by drying, which can be completed by the application of an additional heat process or by natural sunlight. This process could lead to the reduction of the heat load for the evaporation of the water content from the reactor. For commercial purposes, the evaporated water obtained during the drying process of biomass can be sold as steam to support the pyrolysis plant financially [113]. Torrefaction is another thermal pretreatment technique that has been used for the preliminary treatment of biomass for a fast pyrolysis process; this can be classified as mild pyrolysis, because it is processed at a temperature point below 300 °C [116]. The removal of water content to enhance the grind ability, energy density, hydrophobicity, and bacteria resistance is the main purpose of torrefaction [113]. Some authors stated that the application of torrefaction for biomass treatment caused a reduction in bio-oil yield and has an effect on its properties [115,116,117]. Also, when using torrefaction as a pretreatment process so that the bio-oil quality improved, acidity levels are lowered, and energy density is increased [118].

(d)Biological pretreatment

The use of white-rot fungus as a biological pretreatment of biomass in the pyrolysis process enhanced the process performance in the context of pyrolysis temperature and the decomposition of the lignin element [119,120,121]. Also, it was discovered that the application of fungus as a pretreatment might lead to a reduced activation energy demand and an estimated pyrolysis temperature of around 36 °C for cellulose and hemicellulose [121].

(e)Hybrid pretreatment

The hybrid method of pretreatment for biomass is suitable to achieve good quality and environmentally friendly biofuel, pyrolytic gas, and biochar as the target products from lignocellulose biomass in the pyrolysis processes. An experiment performed by Matsakas et al. [35], which narrated a hybrid organosolv–steam explosion resulted in superior digestibility. The experiment was accomplished by the application of ethanol and H_2_SO_4_ into the softwood (spruce) and hardwood (birch) feedstock; the result demonstrated a significant influence of the method parameters on digestibility. Furthermore, the results show that the method favored the birch sample in the production of methane, when compared to the spruce biomass sample. This experiment concluded that the methane production under this method was higher than the conventional process [35].

Also, Charisteidis et al. [122] carried out a similar experiment using spruce and birch biomass samples, which were isolated by the hybrid organosolv–steam explosion technique. It was accomplished by the fast pyrolysis processes resulting in a high content of oxygen, hydrogen, and carbon, while the sulfur and nitrogen content is lower. However, the spruce and birch lignin isolated by the hybrid organoslov–steam explosion method has a minimum amount of ash (<0.1 wt.%), and also contains less carbohydrate impurities, in the sense that hemicellulose and cellulose were (<2 wt.%) and (<1 wt.%), respectively [122].

Generally, the benefit of organoslov-type lignin characteristics is the considerably low content of sulfur and inorganic ash with regard to their valorization, particularly when compared to the kraft lignin and lignosulphonates methods. For instance, in an experiment performed by [123], two different kraft lignin samples, A and B, were quickly pyrolyzed in a Curie-point pyrolyzer in both the absence and presence of HZSM-5. The result showed that sample A contained significantly more coke and less aromatic hydrocarbons than that of sample B and could also result in a negative effect on bio-oil qualities within the higher sulfur content [123].

## 7. System Performance Considerations between Pyrolysis Reactors

When the characteristics/properties of a given biomass are to be determined, particularly when in terms of temperature requirements and the product quality, there is a need for a detailed understanding of the operating system of pyrolysis reactors. For instance, by subjecting the biomass to pyrolysis, Bridgwater [124] reiterated that an understanding of practices/principles is required for the operating processes to happen, with considerations like the (thermal process) characteristics and technology requirements, product characteristics, and even their economics. Besides the key thermochemical approaches of biomass conversion, namely combustion, gasification, liquefaction, and pyrolysis, Bridgwater and Bridge [125] added that anticipated products can be either primary and/or secondary, largely based on the pyrolysis implementation process, all of which create different (pyrolysis) opportunities, constraints, and requirements. In the course of implementing any given biomass gasification as a project, [126] understood that some background knowledge about the gasifier fuel requirements, gasification process, and installation can be useful in understanding its operating performance. Despite these, it is well established that the pyrolysis type is temperature dependent [127,128]. Cotton residue has slowly been pyrolyzed at 300, 350, 400, and 450 °C, and the yields have been measured (Refer to Table 1). Additionally, the production of bio-oil grew continuously as the temperature climbed from 300 to 400 °C. After a temperature rise to 450 °C, the bio-oil output declined to 36.40 wt.%. The gas production grew continuously, as the temperature climbed from 300 to 400 °C. However, secondary cracking was also noted, because the yields of bio-oil were declining as the gas yield increased [128]. As the pyrolysis temperature rises, the amount of char produced by the pyrolysis of shell samples decreases. Between 650 and 800 K, the peak of the liquid yields were recorded. As a result, it seems that pyrolysis temperature affects the char yield and chemical composition. A stronger correlation was found between pyrolysis temperatures and the char components and the higher heating values (HHVs) of shell fuels. Additionally, a highly significant linear correlation was discovered between the pyrolysis temperature of the fuel, HHV, and the fixed carbon content of the char [129].

Temperature has been shown by López and colleagues [130] to have a substantial effect on the characteristics of pyrolysis liquids and, to a lesser degree, both gases and solids. At the lowest measured temperature of 460 °C, a high percentage of highly viscous liquids with a high amount of long hydrocarbon chains are formed, whereas at the maximum evaluated temperature of 600 °C, a low percentage of liquids with a large concentration of aromatics are created [130]. These findings demonstrated that the yield and the quality of biochar are primarily influenced by the temperature applied, with pyrolysis at 600 °C producing biochar with higher fixed carbon (80.70%), carbon (73.75%), higher heating value (30.27 MJ/kg), and lower volatile matter content (9.80%) than the original feedstock, safflower seed press cake (SPC) [131]. PyGC-MS was used to examine how the pyrolysis products of two types of lignin—Asian and Alcell lignin—reacted with temperature. For each type of lignin, 50 or so compounds were discovered and measured over a 400–800 °C temperature range. At 600 °C, both lignins generated the largest production of phenolics, 17.2 wt.% for Alcell lignin and 15.5% for Asian lignin. A phenolic compound’s average yield was less than 1%, while 5-hydroxyvanillin had the greatest output for Alcell lignin (4.29 wt.% on dry ash-free lignin), and 2-methoxy-4-vinylphenol had the best yield for Asian lignin (4.15 wt.% on dry ash-free lignin) [107]. The pyrolysis of poplar wood was thoroughly explored at various reaction temperatures (400, 450, 500, 550, and 600 °C) and heating rates (10–50 °C/min). At the working conditions of 600 °C and 30 °C/min, 600 °C and 50 °C/min, and 550 °C and 50 °C/min, respectively, the BET surface area of biochar, the HHV of non-condensable gas, and bio-oil all obtained maximum values of 411.06 m^2^/g, 14.56 MJ/m^3^, and 14.39 MJ/kg. At 500 °C and greater heating rates, it was possible to achieve a high energy and mass yield of bio-oil, but both lower process temperatures and heating rates lead to a higher mass output and energy output of biochar. Higher pyrolysis temperature and heating rate, on the other hand, lead to a greater non-condensable gas mass production and energy yield. In general, the pyrolysis temperature had a greater influence on the product qualities than the heating rate [132].

Zhang and colleagues studied the yield and physicochemical characteristics of biochar by producing biochar from four feedstocks (wheat straw, corn straw, rape straw, and rice straw) pyrolyzed at 300, 400, 500, and 600 °C for 1 h, respectively. The findings demonstrated that all biochar yields decreased steadily over 400 °C with increasing temperature during the pyrolysis [40]. Due to its higher ash content, biochar made from rice straw had a higher yield advantage. The properties of biochar are significantly impacted by the pyrolysis temperature; these effects can be seen in the negative relationships between H, O, H/C, O/C, (O N)/C, and the functional groups, and the positive relationships between C, ash, pH, electrical conductivity, and surface roughness. Greater pyrolysis temperatures aided in the production of a more resistant constitution and crystal structure, allowing it to be used as a material [133]; this was based on the principle that reactors have been classified [107]. Biomass is composed of hemicellulose, cellulose, lignin, and trace quantities of other organic components, which all pyrolyze or decompose in various ways and at different rates. Lignin’s apparent thermal resilience during pyrolysis is owing to the fact that it decomposes across a greater range of temperatures than cellulose and hemicellulose, which breakdown relatively quickly over smaller temperature ranges. The temperature, rate, and pressure of the reactor (used for pyrolysis) determine how quickly and how thoroughly each of these components decompose. The amount of the secondary reaction (and hence the product yields) of those products is determined by the time–temperature history that the gas/vapor products are subjected to before collection, which includes the impact of the reactor setup [124].

In a study conducted by Yufeng and colleagues, the technology used in China’s landfills, incinerators, and other methods of disposing of municipal solid waste were all examined. In China, a new device has been created for waste disposal that is based on the traditional pyrolysis principle. In China, where waste is not sorted, it is particularly helpful. By adjusting the residence time and temperature, the experiment demonstrates that the concentration of dioxins satisfies the emission standard of 0.1 ng-TE/N m3. As little as 5–7 percent of the total weight of the waste is expulsive solid. The treatment process also produced a significant amount of fire gas [134]. In addition to the term “pyrolysis”, which relates to the process of decomposing biomass using heat and no oxygen to produce charcoal, liquid, and gaseous products, the term “pyrolysis” also refers to three subclasses of the process: conventional pyrolysis, fast pyrolysis, and ash pyrolysis. Hemicelluloses decompose at temperatures between 470 and 530 K, cellulose follows at 510 to 620 K, and lignin is the last material to pyrolyze at 550 to 770 K. To increase the output of liquid products generated by biomass pyrolysis, a low temperature, high heating rate, and brief gas residence period process would be required [135]. There are other differences between the operational methods of pyrolysis reactors, namely snapshots of the single-operated pyrolysis method; snapshots of combined thermal conversion treatments and analytical methods; and other miscellaneous/pyrolysis-mimicking operations. Upon a thorough check of the relevant literature, we authors observed that there are an array of pyrolysis reactors that have been used across various studies. Additionally, being the heart of any pyrolysis process, authors like Jahirul et al. [24] understood reactors to be considerable for research interests and sustainable routes for diverse biomass innovation/development. To improve the pyrolysis process, operational aspects like heating/temperature rates and (product) residence times are among the essentials that have to be considered [24]. A schematic representation of the pyrolysis temperature reactor increases based on (1) single, (2) combined, and (3) miscellaneous operating systems, reflecting a distinct categorization of various reactors, is shown in Figure 4. The essence of creating the abovementioned operating systems is to evaluate such pyrolysis reactors, specific to which context the reactors were used, and also which condition had to be fulfilled for a specific reactor to perform. Subsequently, herein, we discuss the above-mentioned operating systems in greater detail, largely in the context of pyrolysis temperature reactor increases.

### 7.1. Snapshots of The Single-Operated Pyrolysis Method

The single-operated pyrolysis includes a fixed bed, thermogravimetric analysis (TGA), an automatic methane potential test system, vertical dual-bed tubular quartz, tubular quartz, extrained flow gasification, a cylindrical reactor, a furnace reactor, a drop-tube furnace, a rotary kiln, wire mesh, ablative, a fluidized bed, semi-batch vertical, and hydrothermal carbonization (HTC). Each of these is succinctly discussed below:(a)Fluidized bed reactor

The quality of renewable jet fuel-like iso-alkanes, especially those in the products, has been considered necessary for improvement. This is what Chen and colleagues envisaged, when they used a fluidized bed reactor connected to a hydro-conversion system for the processing of rice husks. At a temperature between 320 and 400 °C, the hydro-cracking and isomerization processes were carried out. These authors described their operation process as fast pyrolysis [26]. There is a paucity of literature regarding catalysts that are able to promote lignin depolymerization. On this basis, a continuous fluidized bed reactor was utilized to investigate the ability of ferrous, ammonium, and magnesium cations in combination with sulfate anions, directly aimed to prevent the agglomeration, and at the same time, to promote the formation of sugar during the herbaceous biomass pyrolysis. The char cyclones are subjected to a high temperature of 500 °C in a heat tape, which signals a fast pyrolysis process [81].

(b)Fixed bed

Sieradzka et al. [38] integrated the capturing of CO_2_ with biomass thermochemical conversion pyrolysis and used a fixed bed in the process. In this instance, the effects of the pyrolysis temperature (500, 600, and 700 °C) and CaO sorbent addition were evaluated, considering both chemical and physical properties; this aimed to obtain the char and syngas [38]. Given the above temperature range, this study showed an example of fast pyrolysis conditions, where the increasing temperature in syngas brought about changes in solid products, with a decreased CO_2_ concentration. Su and colleagues [34], in their attempt to overcome the rice husk defects, so as to provide renewable energy/materials via the pyrolysis poly-generation method, used a fixed-bed reactor for char and bio-oil as the target products. These authors deemed rice husks as a promising target product with less emissions. From the activation process that operated at 500 °C for 90 min under N_2_ protection, the fast-pyrolysis method was obviously supported by catalytic means, which employed Na_2_CO_3_ for an enhanced product quality [34].

Given the limited knowledge of the characteristic features of chars produced from the co-pyrolysis of cellulose and lignin in Chua et al. [75], who utilized a drop-tube/fixed-bed quartz reactor with pulsed feeding at temperatures under 350 °C, it was specifically studied how cellulose, and lignin interacted during fast pyrolysis. These authors were able to better the understanding regarding the fundamental pyrolysis mechanisms of lignocellulosic biomass. The release of volatiles from cellulose and lignin was enhanced at temperatures below 300 °C, due to the decline of lignin functional groups and sugar structures within the char. The co-pyrolysis of cellulose and lignin, however, increased the char yield to about 300 °C [75].

Given the existing uncertainties regarding the fundamentals of the levoglucosan (LGA) conversion to the levoglucosenone (LGO) reaction system, the liquid phase transformation of anhydrosugars over solid acid catalysts was investigated. To achieve this, an updraft fixed-bed pyrolyzer was employed and operated at 500 °C, which had LGA with a yield of 38.4%-C and negligible LGO, after which there was a reaction and product analysis that involved high-performance liquid chromatography. As LGO is produced with a yield of up to 32.3%-C, a portion of heavier saccharides would contribute as a source of LGO without impeding the conversion of LGA [11]. Given the need to reduce the fuel load in the forests as is consistent with the national biomass valorization policies, the workers looked into the yield and properties of charcoal produced from ten common Southern Europe wood types, subject to operational conditions that were deemed relevant for biomass carbonization technologies. Particularly, a fixed-bed reactor was used, which allowed large fuel particles to be subjected to different heating rates of between 0.1 and 5 °C/min, with final temperatures of between 300 and 450 °C. Fast pyrolysis was the operation process that best fit this temperature range and the use of a fixed-bed reactor [50].

In order to comprehend the cellulose pyrolysis mechanisms and the development of its biochar structures, Zhang et al. [43] used a fixed-bed reactor at temperatures between of 200 and 800 °C when studying the product output at each increment of 50 °C, by characterizing the emergent gaseous products, liquid products, and bio-char. This temperature range of 200–800 °C, together with the fixed-bed reactor type, confirmed that the operation process had moved from slow to fast pyrolysis. To determine the worth of the apparent isoconversional activation energy profiles, *Sobek* and Werle [83] applied fixed-bed-based solar pyrolysis to three waste biomass types: waste straw (WS), sewage sludge (SS), and waste wood (WW). The aim primarily was to study the heating behavior, the products quality, and the yields. Specifically, the temperature range of 0–1200 °C showed the operation process had clearly moved from slow to fast pyrolysis. Considering the excellent results that the chemical activation of bio-char had generated from the rice husk pyrolysis, a horizontal oven with a quartz tube was used to chemically activate the rice husk biochar, using a K*_2_*CO*_3_* activating agent, while also using solid-phase extraction (SPE) in order to remove any potentially harmful organic compounds, specifically from the bio-oil aqueous phase. The operation temperature of the tubular fixed-bed oven ranged between 430 and 620 °C, whereas that of the horizontal oven containing a quartz tube that did the activation ranged 800–900 °C and was maintained for 2 h, which clearly demonstrates fast pyrolysis [37].

(c)Thermogravimetric analysis (TGA)

For the better utilization of biomass fuel, Jia et al. [44] determined the main chemical components of pellet types (namely Chinese fir, Masson pine, Slash pine, and Poplar) using the thermogravimetric analysis method (and Coats–Redfern method), which involved a kinetic analysis. The experimental operation utilized raised temperatures of up to 845 and 900 °C, respectively, subjected to a nitrogen and oxygen atmosphere at 5 mL/min. These temperatures attests to the fact that this operation process was an example of fast pyrolysis [44]. To determine the variations in the pyrolysis properties of different biomass types, H. Chen et al. [64] used the thermogravimetric analysis method to determine the pyrolysis and to categorize the 20 types of biomass in three groups (stalk, wood, and shell type). The pyrolysis characteristics were explored based on how the biomass types and mechanisms were affected. With 60 mL/min of pure nitrogen purging, the sample heating rate is 15 °C/min from 30 °C to 900 °C; this operation process attests to the combination of slow and fast pyrolysis [64].

Some concerns were associated with the reliability of both the experimental and modeling aspects of previously conducted TGA-pyrolysis studies. This was understood by [33], when they performed TGA pyrolysis investigations on pure cellulose and beech wood, taken at several heating rates, incorporating holding times of 10–15 min, with temperatures of between 150 °C and 500 °C, and at the peak which allowed for char yield. The idea behind their study was to improve TGA for biomass pyrolysis specific to the consistency of kinetic analysis and data acquisition. Given these temperatures of between 150 °C and 500 °C, the operation process can be considered an example of slow to fast pyrolysis. To analytically assess the energy characteristics of torrefied biomass under specified pyrolysis conditions involving typical woody and herbaceous biomass, based on isothermal pyrolysis kinetics, TGA was used to investigate the assessment methods of the HHV and mass yield of torrefied biomass on three biomass species: (i) hardwood; castanopsis, (ii) herbaceous biomass; rice straw, and (iii) softwood; Japanese cedar. Overall, the temperature ranged from 105 °C to a predetermined torrefaction temperature (230–310 °C) at 20 °C/min, some of which involved a predetermined residence time (0.5–4 h). This temperature range of between 105–310 °C typified that of slow pyrolysis [56].

There is a paucity of information regarding the pyrolysis kinetics of orange and potato peels. TGA was applied to ascertain the kinetic parameters, which included (a) a pre-exponential factor, (b) activation energy, and (c) a reaction order, which involved either model-fitting or model-free methods, both differential and integral ones. Using the heating rates of between 2–15 °C/min and the TGA temperatures ranging from ambient to 650 °C, this operation process can be typified as moving from slow to fast pyrolysis [55]. To consider the potential secondary gas–fuel reactions, particularly when applying large-scale pyrolysis processes, in a study that used congruent–mass thermogravimetric analysis and conventional thermogravimetric analysis methods to pyrolyze individual coal (Datong bituminous coal) or biomass (bamboo and wheat straw) samples, using the same operating conditions, specifically heating temperatures from room to 900 °C. This temperature range typified the operation process that moved from slow to fast pyrolysis [47].

Given that different heating rates could change the reaction kinetics, many model-fitting techniques seem to be less effective for the pyrolysis of biomass. Fakayode et al. [68] used TGA to examine the energy and higher heating value (HHV) of ultrasound-assisted deep eutectic solvent pretreated watermelon rind biomass (WMR). The TGA heated at rates of 5, 10, and 20 °C/min from 35 to 1000 °C. In particular, the heating temperature was held at 1000 °C, until attaining steady conditions that detected no further mass loss [68]. Overall, these temperature ranges between 35 to 1000 °C typifies moving from slow to fast pyrolysis. Considering the difficulties associated with highly complex models for practical application purposes, particularly in evaluating char preparation, Fermoso et al. [76] utilized a pressurized thermogravimetric analyzer (PTGA), using CO_2_ as a gasifying agent under isothermal conditions at different temperatures (750–900 °C) at 40 °C/min, and pyrolyzed the chars at temperatures between 1000 and 1400 °C, with residence times for the particles of approximately 7 s. Overall, these temperature ranges between 750 to 1000 °C typify gasification/fast pyrolysis.

Providing a theoretical basis to optimize a pyrolysis process that effectively utilizes corn straw resources is very important. This is what Chen et al. [72] understood when they used a thermogravimetric analyzer from room temperature to 700 °C, under five heating rates (10 °C/min, 20 °C/min, 30 °C/min, 40 °C/min, and 50 °C/min) on HCl-washed corn straw, and then determined the biomass and pyrolysis of the material. Overall, these temperatures from room temperature to 700 °C would typify a movement from slow to fast pyrolysis. Considering the dependency of the yield and quality of bio-oil that emerged from these pyrolysis processes on several factors, which can involve biomass property, operating conditions, pyrolysis types, and reactor types, is incredibly important. Shrivastava et al. used thermogravimetric analysis (TGA), largely involving a range of 200–450 °C, to produce bio-oil via pyrolysis processes, and subsequently determined the potentiality of oil palm biomass, as well as oil palm fronds (OPF), an oil palm decanter (DC), oil palm trunk (OPT), and oil palm root (OPR). The 200–450 °C temperature range of the operating process suggests slow pyrolysis [79].

To know the kinetic parameters and energy properties of a given biowaste at thermal decomposition, Noszczyk et al. [42] used a thermogravimetric analyzer—the Pyrolysis Biomass Gasifier—to study the energy and kinetic parameters of peanut, hazelnut, pistachio, and walnut shells. The TGA operated at heating rates of 5, 10, 20 °C/min, from 30 °C to 900 °C, which typifies slow to fast pyrolysis. Due to the lack of comparative pyrolysis investigations on different corks that would enable an understanding of its behavior and how it can be used in the reactor/process design for industrial biochar/bio-oil, [69] used the (TGA) analysis to evaluate the different characteristics of corks by pyrolysis behavior to target scaling up, both in the valorization strengthening of these materials, and the integration in thermochemical platforms. The TGA operated isothermally from 30 °C for 10 min, linearly heating up stepwise until 800 °C, with varying heating rates (10, 2,0 and 50 °C/min). This operating process typifies movement from slow to fast pyrolysis [42].

Given the paucity of the understanding of the thermal behavior of specific biomass processing, Gözke and Açıkalın used thermogravimetric analysis to determine the (pyrolysis) properties and kinetics of sour cherry flesh and stalk. The TGA temperatures were set from ambient to a maximum of 1000 °C at 5, 10, 20, 30, and 40 °C/min. This operating process typifies movement from slow to fast pyrolysis [58]. Moreover, there is a scarcity of data on the kinetics of exhausted coffee residue (ECR) and coffee husks (CHs). To supplement existing information, Mukherjee et al. used TGA analyses to study the pyrolysis kinetics and thermal degradation of ECR and CHs in an inert atmosphere. The operating temperature program ranged between 25 and 800 °C, with heating rates ranging from 5–20 °C/min, with an interval of 5 °C/min, which typified movement from slow to fast pyrolysis [59].

Singh et al. [60] used thermogravimetric analysis (TGA) to study the thermal degradation of banana leaves waste based on the kinetic triplet (pre-exponential variable, activation energy, and reaction model) at 10, 20, and 30 °C/min. The operating temperature ranged from ambient to 900 °C, which typified movement from slow to fast pyrolysis. Because of the little information regarding thermo-kinetic investigations involving the pyrolysis of bacterial biomass (BB), the bioenergy capability of a subset of biological waste from butanol, acetic acid, ethanol, and lactic acid producing facilities was tested using TGA analysis. Together with (three) heating rates of 10, 20, and 30 °C/min, the operating temperature ranged from room 25 °C to 700 °C, which typified movement from slow to fast pyrolysis [61].

As hydrochar-derived biomass via pyrolysis has strongly depended on its origin, Magdziarz et al. [82] used hydrothermal carbonization (HTC) and a pyrolyzer (Pyroprobe model 5200, CDS Analytical) with GC-MS and thermogravimetric analysis (TGA) to determine the energy potential of hydrochars derived from energy crop (Virginia mallow), agriculture biomass (straw), and wood biomass (pine). The HTC operation process involved 220 °C and 4 h temperature and residence time, respectively. The TGA, with a heating rate of 10 °C/min, had a temperature range from ambient to 700 °C. However, the Py-GC/MS had a temperature range from 40 to 600 °C. Overall, the operation process appears to be a combination of slow and rapid pyrolysis. The pyrolysis behavior of *Phragmites hirsuta* is seldom studied, especially with respect to the pyrolysis mechanism. Therefore, in Liu et al. [62], *Phragmites hirsuta* root, stem, and leaves were subjected to a thermogravimetric analysis in order to ascertain their pyrolysis behavior and kinetic properties as a potential source of bioenergy. The thermogravimetric analyzer was able to operate between 30 and 900 °C at different heating rates (10–50 °C/min). The operation process typifies a movement from slow to fast pyrolysis [62].

(d)Other peculiar pressure gas-based reactors

The pressurized entrained-flow gasification (PEFG) of straw biomass as a potentially sustainable and commercially viable process to produce fuels, and the understanding of the fractionation of inorganic constituents with respect to gasifier conditions and various straw compositions are two areas of interest. Mielke et al. used PEFG to identify the relevant fractionation processes that are dependent on ash composition, employing predicting slag composition and viscosity models based on the ash composition of the fuel and the process parameters. Pressurized entrained-flow gasification (PEFG) operated at 1400 °C, with varying retention times from 10 s to 50 s, which typified a fast pyrolysis process [74].

There was a paucity of detailed and complex analytical exposure to ablative fast pyrolyzed (AFP) bio-oils, especially with insight into prevailing differences. It was this gap that made [6] use a 5 kg/h unit ablative fast pyrolysis (AFP) lab-scale reactor to evaluate the biomass type, properties, and composition of bio-oils that have been produced from poplar wood and beech, miscanthus, and straw. The pyrolysis operated at a 550 °C constant temperature. This operation process signaled fast pyrolysis. Elsewhere, because eco-social business models that are more cascading and have circular-based environmental, social, and economic benefits within the food waste sector are needed, Matrapazi and Zabaniotou applied wire mesh captive sample type reactors on spent coffee grounds in a large-scale slow pyrolysis [48].

### 7.2. Snapshots of Combined Thermal Conversion Treatment and Analytical Methods

(a)Fixed bed with torrefaction

In studying the evolution of functional groups during the wet torrefaction process, Wang et al. [52] conducted a comparative analysis of torrefied corn stalk using a vertical fixed-bed, investigating how biomass pyrolysis polygeneration takes place under optimal conditions. It should be noted that the reactor heating was at a torrefaction temperature (200–290 °C), then the sample (5 g) was swiftly placed in the reactor center. These workers found biochar yield after wet torrefied less than dry torrefied, with the upgraded biochar quality given the high ash removal. This was followed by pyrolysis properties of torrefied samples in terms of bio-char, pyrolytic gas, bio-oil, and yield distribution. This exemplified a combination of slow (torrefaction) and fast (vertical fixed-bed reactor) pyrolysis operation.

(b)Thermogravimetric analysis–Fourier transform infrared (TGA–FTIR)

Considering the processing of mixed solid waste that can adopt a two-stage solid prototype, Serio and Wójtowicz used TGA–FTIR analysis with the FTIR analysis of the evolved gases system to determine how feasible it is that an advanced methodology can be developed to evaluate the biomass materials. To actualize the pyrolysis process, the TGA–FTIR operated from 150 (for 3–4 min) to 900 °C, which typifies the move from slow to fast pyrolysis [41].

Researchers have pursued more information due to a requirement for more knowledge on the kinetic characteristics of biomass with complicated thermal properties. Da Silva et al. [63] analyzed the kinetic parameters using a thermogravimetric analyzer (activation energy, frequency factor, and reaction model) to investigate the pyrolysis of biomass with complex thermal behavior, including cashew nut shell waste (CSW) and sugarcane bagasse waste (SBW). Five different heating rates were used during this operation, ranging from room temperature to 1073 K (about 800 °C), showing the progression from slow to fast pyrolysis. Furthermore, particularly from the agro-industry standpoint, the green corn husks as biomass, via pyrolysis, can be an alternative energy source [63]. It is on this premise that Reinehr and colleagues [70] used a thermogravimetric analyzer, through pyrolysis reaction kinetics, and were able to perform the analysis of green corn husk properties and characterizations, so as to find the thermokinetic conversion parameters. The TGA operated at 30 to 900 °C, with heating rates of 5, 10, 15, and 20 °C/min, which depicts movement from slow to fast pyrolysis.

(c)Thermogravimetric analysis (TGA) and differential thermo-gravimetry (DTG)

To identify the future biofuel potential use of corncob and eucalypts, Kumar and colleagues investigated the thermal degradation, kinetic parameters/properties, and the deconvolution of biomass/combustion characteristics, after having them subjected to differential thermo-gravimetry (DTG) and thermogravimetric analysis (TGA). The pyrolysis temperature was found to have attained up to 1000 °C, which demonstrates that this operation process was fast pyrolysis [32].

(d)TGA–FTIR and Py-GC/MS

In order to understand the volatile compositions and their formation pathways/kinetics during biomass pyrolysis that help in regulating the target products’ quality, Tian et al. applied two pyrolysis stages that coupled real-time volatile monitoring techniques (Py-GC/TOF-MS and (TGA–FTIR) to rice husks that were subjected to three different heating rates (10, 20, and 30 °C min^−1^), starting from room to 800 °C. In particular, the temperatures from room to 800 °C demonstrated that this operation process moved from slow to fast pyrolysis [31].

Given that information on comparative studies on two-step pyrolysis (TSP) of different lignocellulosic biomass and the effects of components on TSP were scant, Zhang et al. [40] applied TGA–FTIR and Py-GC/MS in studying the effects of TSP on lignocellulosic biomass, by comparing corncob (CC), cotton stalk (CS), walnut shell (WS), and their acid-washed samples (ACC, ACS, and AWS). The TGA–FTIR, at a heating rate of 20 °C/min, operated from room temperature to 750 °C in order to realize the vapors, whereas Py-GC/MS operated a two-step process, first conducted at 400 °C for 20 s, and second, at 650 °C with a residence time of 20 s, to realize the volatiles. Given these 650–750 °C temperature ranges, the operation process can be considered as a fast pyrolysis.

(e)Thermogravimetric analysis (TGA) and Pyrolysis-GC/MS

To better understand the effects and importance of parameters (like biomass composition, particle size, shape, residence time, and heating rate) on the devolatilization and bio-oil composition kinetics for a successful process scale-up, Vinu [54] employed Py-GC/MS and TGA to pyrolyze mixed wood sawdust (MWSD) of eight different particle sizes (26.5–925 μm) at different heating rates of very slow (<3 °C/min), slow (5–20 °C/min), medium (50–100 °C/min), and fast (10,000 °C/s). Specifically, the TGA temperature ranged between 25 and 900 °C, whereas the filament temperature of Py-GC/MS was set at 600 °C and maintained for a period of 30 s. Considering the temperature ranges of 25 and 900 °C, the operation process can be considered to have moved from slow to fast pyrolysis.

Because of the fact that investigations into the correlation of aldehydes, furans, and ketones with carbonyl groups in bio-oil with holocellulose appear scantly, Y. Liu et al. [49] used a Pyroprobe 6200 pyrolizer (Py-GC/MS) and TGA to study the pyrolysis behaviors of nine biomass-derived holocelluloses (from seven agricultural and two forestry residues). The TGA operated from room temperature to 800 °C at 40 °C/min, whereas the pyrolysis-GC/MS had its platinum spiral coil’s heating rate of 10,000 °C/s operating heated from 50 °C to 550 °C, maintained for 15 s. The process from slow to fast pyrolysis is typified by the entire temperature range between room temperature and 800 °C. In order to produce and use syngas, bio-oils, and value-added chemicals, while reducing waste stream and greenhouse gas emissions, it is possible to use TGA–FTIR and Py-GC/MS analyses. This was the foundation [53] used in their combination of TGA, FTIR, and Py-GC/MS analyses used to quantify the bioenergy and by-product outputs at different heating rates. The TGA operated from room temperature to 1000 °C at 5, 10, 20, and 40 K/min heating rates, which typifies movement from slow to fast pyrolysis.

(f)Other thermogravimetric analysis combinations

Besides conventional pyrolysis processes used to bring about thermally unstable oxygenated bio-oils, carbon-rich solids in biomass pyrolysis (i.e., biochar) remain the economical choice for catalytic applications. Hao et al. [39] used a thermogravimetric analysis for pyrolysis at temperatures of 20–750 °C and used a vertical dual-bed tubular quartz reactor at a temperature of 300 °C for 2 h in order to study how temperature and mixing ratio affect the straw (RS) and *Ulva prolifera* macroalgae (UPM) product distribution by catalytic and non-catalytic co-pyrolysis. Generally, the operation temperature range of 20–750 °C demonstrated slow to fast pyrolysis. Elsewhere, there is a paucity of relevant data regarding how operating pressure influences the thermal effects of the pyrolysis process, and that is why Basile and colleaguesused the thermogravimetric analyzer (TGA) at a heating rate of 10 °C/min and a final temperature of 950 °C, whereas the differential scanning calorimetry (DSC) for pressures at 0.1, 0.5, 1, 2, and 4 MPa, the constant heating rate was 10 °C/min, then arriving at the final temperature of 550 °C. The temperature of the operating process suggests fast pyrolysis [78].

(g)Analytical pyroprobe^®^ reactor and Pyrolysis-GC/MS

Given the differences in the lower and faster heating rate conditions, which obtain kinetic parameter validation as the requirement for weight loss profile data to be reliable, Ojha et al. [57] used an analytical pyroprobe^®^ reactor, first with FTIR, to study the isothermal mass loss of biomass, and then, combined with gas chromatograph/mass spectrometer (Py-GCMS) to look into the kinetics of fast pyrolysis of three lignocellulosic biomasses, i.e.,. empty fruit bunch (EFB), pinewood (PW), and rice straw (RS). These authors used a Pyroprobe^®^ reactor with FTIR which operated at 400, 450, 500, 550, 600, 650, and 700 °C, using hold times of 2, 4, 6, 8, 10, 15, 20, 30, and 60 s, whereas the Py-GCMS used temperatures (400, 500, 650, and 800 °C) and held for 30 s. Overall, these temperature ranges between 400–800 °C typify fast pyrolysis [57].

### 7.3. Other Miscellaneous/Pyrolysis-Mimicking Operations

(a)Hybrid organosolv–steam explosion reactor

In order to combine the fractionation ability of the organosolv system to physically reduce the size of the biomass during the steam explosion, and at the same time, to pretreat and fractionate the birch and spruce biomass, Matsakas et al. [35] studied how the digestibility was influenced by the different process parameters of the hybrid method. To achieve this, these workers used both a hybrid organosolv–steam explosion reactor and Automatic Methane Potential Test System II, subjected to temperatures of 200 °C and 55 °C for up to 18 days, respectively. Even though a reactor was used, and despite the temperature of 200 °C, we opine it to be a biological process, given the nature of this study. A performance evaluation of the novel process steps for converting biomass should take into account the high fractionation efficiency of organosolv pretreatment. Mesfun et al. [106] utilized a hybrid organic solvent and steam explosion pretreatment technique to separate lignocellulosic biomass onto streams rich in cellulose, hemicellulose, and lignin in order to determine how well it would perform in a biorefinery setup. With a holding period of 15 min, the used hybrid organic solvent and steam explosion pretreatment reactor were ran at 200 °C, which typifies a slow type of pyrolysis.

(b)Greenfield Eco. Pvt. Ltd./Cylindrical furnace reactor

Consequently, the production of biochar from invasive weed mesquite biomass could benefit waste management, prevent CO_2_ emissions, and soil amendment could also aid in carbon sequestration and soil improvement. Hussain et al. [77] used the Greenfield Eco. Pvt. Ltd. pyrolysis instrument to determine the impact of biochar on the soil’s hydraulic characteristics, thereby assessing its suitability for farming. The temperature was set at 500 °C, which typifies slow pyrolysis, as stated by the authors. To contribute to the quest to discover various alternate fuels, like the depletion of fossil fuels and environmental impacts due to emissions of IC engines, Thamizhvel et al. [45] developed a bio-fuel using various techniques from feedstock, and subsequently conducted an analysis on its properties. This pyrolysis used a cylindrical reactor placed in a furnace, the temperature was set to 600 °C and connected to a gasifier, which the operation typifies as a fast pyrolysis/gasification condition, as stated by the authors.

(c)Hydrothermal carbonization (HTC)

Because different biomass components would bring about changes in the thermal conversion, which would then influence the physical/chemical properties of the char, Xu et al. [65] used hydrothermal carbonization (HTC) (temperature 220 °C for 4 h with 2.0–2.5 MPa pressure), combined with a stainless steel cylinder reactor (having a temperature between 300–800 °C at 10 °C/min), to pyrolyze biochar and compare with hydrochar, with the operation process being implemented from ambient to the desired temperature, which typifies slow pyrolysis. The nature of hydrochar is guaranteed with high carbon content and porosity. Additionally, both hydrothermal carbonization and pyrolysis can deliver more porous materials with a higher carbon content. Bahcivanji et al. [67] opined this when they applied hydrothermal carbonization (HTC) to waste biomass (WB) feedstock, eventually pyrolyzing the samples at temperature ranges between 350 and 550 °C, across 1, 3, and 5 h periods, which signals a slow system approach. Moreover, there are numerous thermal conversion reactors used to conduct any pyrolysis based-study to determine and investigate any given feedstock properties and the target products, namely muffle furnace-based pyrolysis, a tubular quartz reactor, a drop tube furnace (DTF), a semi-batch vertical reactor, a plug flow reactor (PFR), and a continuous stirred tank reactor (CSTR), via simulation.

(d)Other pyrolysis instruments

There are other pyrolysis instrument reported, namely muffle furnace-based pyrolysis; a tubular quartz reactor; a drop tube furnace (DTF); a semi-batch vertical reactor; a continuous stirred tank reactor (CSTR), and a plug flow reactor (PFR) via simulation.

In order to understand the temporal changes associated with particulate matter (PM) characteristic/properties and its emission during combustion, Itoh et al. [136] used a muffle furnace to evaluate the impacts of the operating temperature on dairy cattle manure and wood shavings. They pyrolyzed the samples at temperatures of 200, 300, 400, or 500 °C for 1 h. The operating process clearly demonstrates the combination of slow and fast pyrolysis. Because the quantification of anhydro-sugars appears challenging and its conventional analysis requires pretreatment, Téllez et al. [36] employed a tubular quartz reactor in order to evaluate the content of Levoglucosan (LG) in the bio-oils from pyrolyzed (hydrochloric acid-treated and untreated) rice husks. The temperature operation fell between 300 and 700 °C, which demonstrated both slow and fast pyrolysis.

There was the need to provide additional information and understanding regarding fly ash formation during bio-oil/biochar combustion, as well as to elucidate the differences and similarities when compared to another relative raw biomass. Based on this, Johansson et al. [80] utilized a drop tube furnace (DTF) with a maximum process temperature (of 1400 °C) to pyrolyze five different biomass powders (forest residue, stem wood, willow, bark, and reed canary grass), in order to ascertain the formation of fly ash during suspension combustion and the corresponding products. This maximum process temperature (1400 °C) signals the fast pyrolysis of the biochar and bio-oil of the powders. There is believed to be a high potential of biowaste application as the energy source in Poland; this is in-line with the growing world demand for the pyrolysis of waste materials. It was based on this that Mlonka-Mędrala et al. [51] used a semi-batch vertical reactor at 300–600 °C on oat straw in order to examine its potential as a technology for managing biomass waste. This temperature range, 300–600 °C, signals from a slow to a fast form of pyrolysis. SuperPro Designer (SPD) has been poised to perform modeling and simulation tasks that engage various biomass conversion processes. Pang et al. [71] simulated a pyrolysis process that employed a CSTR for the primary decomposition of biomass, and a PFR to model the remaining fragmentation of unreacted components that would form bio-char, gas, and oil. It was shown that both reactors were set to operate at 550 °C and 1 atm in order to simulate the actual biomass pyrolysis, which signals a fast process.

## 8. Differentiating between the Reactor and Operation Parameters Involved in Thermal Conversion Processes

It is important to understand the operation (parameters), especially where the pyrolysis reaction takes place, since the reactor is one of the most significant elements determining the yield of the fast pyrolysis product. This would make the target product(s) from feedstock associate with the heating rate of the system, as well as the heat transfer method. Notably, reactor types and operating methods play major role in pyrolytic products’ quality, yield, and cost efficiency, as shown in Table 4. Many researchers show fluidized beds (bubbling and circulating) as advantageous and more lucrative, in terms of product output [127,137,138]. Examples of pyrolysis reactors include rotating cone reactors, fluidized beds, ablatives, circulating fluid beds, and auger reactors [5,138], as shown in Figure 5.

One of the criteria influencing the quick pyrolysis yield of products is the reactor. There are several varieties that differ in their working process (Table 4), which influence the quality of the products, energy demand, reactor capacity, energy transfer, particle size, and gas emission. Fluidized beds (bubbling and circulating) have been found to be more profitable and suitable, in terms of product yield and quality in numerous studies that have looked into the matter [137]. According to Peacocke et al. [139] an experiment was performed, aiming to compare the fluid bed and ablative pyrolysis reactors under the same operating parameters, including the process temperature value. The results showed that, in both reactors, char yields increased above 515 °C. The ablative produced higher volatile content char than the fluid bed results, where the chars decreased rapidly. However, in the case of liquid yields, the results are similarly in the range of 11–16 wt.%, and gas yields were recorded lower in the content of the ablative reactor, indicating a less severe environment for the vapor [139]. The shift in the chemistry of the gaseous products in the fluidized bed with temperature indicates that secondary vapor phase cracking in the fluidized bed is more prevalent when compared to the ablative process [139].

The thermal conversion-based process normally begins at temperatures between 200 and 300 °C, while volatiles are quickly liberated in the absence of oxygen at temperatures between 750 and 800 °C [140,141]. Generally, this process comprises five (diverse) process types, namely pyrolysis, hydrothermal carbonization, gasification, combustion, and torrefaction [1]. Each process gives a different range of products and employs different equipment configurations, operating in various modes. The main characteristics of these processes are described in Figure 6, including the product properties and yield [1,39,46,140,142].

### 8.1. Snapshots of Single-Operated Pyrolysis Method

When temperature increases in the pyrolysis process, it would lead to increasing the gas production yield and decreasing the char yield. A maximum bio-oil yield tends to be achieved at a range of 400–600 °C pyrolysis process temperature levels, and at a water content decrease, as a result of a higher organic yield [143,144]. However, it depends on the feedstock. It was investigated in many studies that, in the case of wood feedstock, around 500 °C is usually the maximum temperature point [145]. Higher process temperatures leads to a decrease in the hydrogen-to-carbon ratio and the oxygen content. The heat flux and heating rate increase in direct proportion to the environment temperature [145].

Pressure is one of the pyrolysis parameters which impacts the pyrolytic products’ yield and quality. Generally, pyrolysis pressure has a significant effect on the size and the shape of the particles through increasing the proportion of void space, resulting in decreasing the cell wall thickness. Biomass particle swelling occurs at low pressures, and higher pressure pyrolysis leads to larger char particle size and bubble formation, while an increased pyrolytic pressure leads to slight decreases in the total surface area, for instance, 1 bar compared with 20 bars [21,76]. In addition, it was investigated in numerous studies that a raise in the operating pressure led to the decline of the heat requirements of the pyrolysis process, and the high-pressure operating process may lead to a shift in the process heat from an endothermic to an exothermic process [78,144]. For example, Lucia Basile et al. [78] utilized a specially designed experimental configuration method, in which DSC was employed to determine the heat demand of the pyrolysis process at operating pressures ranging from 0.1 to 4 MPa. The results showed that, as the operating pressure was raised, the heat demand declined, and the final char yield improved. The obtained results suggest that there is a competing mechanism between the endothermic reaction of the primary decomposition process, which results in the synthesis of volatiles, and the exothermic vapor–solid contact, which results in the development of secondary char [78].

### 8.2. Considerations of Residence Time and Particle Size

In general, the residence period for rapid pyrolysis is less than 2 s, while for slow and moderate pyrolysis, it is higher [146]. Typically, this decreases the secondary reactions such as thermal cracking, bio-char development, recondensation, and repolymerization, leading to a decline in organic yields while the yield of char and permanent gas increases. Additionally, an experiment was carried out by Xu and Tomita [144] where the effects of residence time and the pyrolysis cracking temperature of volatiles on pyrolytic product yields were determined, ranging from 0.2 to 14 seconds and between 500 to 900°C, respectively. The results showed that as the residence time becomes longer at a given cracking temperature, the tar yield decreased while the yield of gas and light hydrocarbon liquid increased [144].

The majority of pyrolysis operating reactors required small particle sizes in a vertical riser, which allows for high heat transfer rates in the process ranging between 0.5 to 5 mm [146]. The ash content of biomass decreases as the fixed carbon and volatile matter content increase, and vice versa, as the biomass particle size increases, although the size depends on the operating reactor types. In addition, the limited heat transfer between particles as a result of the larger particle size led to relatively higher average activation energies. Therefore, small particle size is an advantage to achieving the pyrolysis process with low energy transfer [54,147].

### 8.3. Considerations of Energy Demand

The energy consumption in pyrolytic operations is one of the factors to consider due to its impact on the yield and quality of pyrolytic products. However, it is dependent on the feedstock qualities and the operating reactor. Heat transfer requirements are crucial for the efficient conversion of biomass and must be fulfilled. Reducing the size of biomass particles can increase heat transfer rates because biomass has poor thermal conductivity. The insulating char layer developed on the surface of biomass during pyrolysis progression also contributes to the heat transfer resistance. The incremental impact of char formation on heat transfer resistance can also be lessened by decreasing the size of the biomass particle [147]. However, size reduction adds to the cost of feedstock preparation because it is an energy-intensive process. Rapid heating rates promote the quick breakdown of biomass, resulting in more gases and less char in the process. Rapid heating also leads to a high production of bio-oil [148].

## 9. Knowledge Gaps and Future Prospects

Based on this review, several interesting gaps in knowledge were identified. Researchers in this field may use this as a foundation for further research. Although a number of technologies and approaches have been investigated for a decade, the attempt to lag/isolate the external part of most thermal technologies has not been explored. However, achieving high product quality and yield remains one of the technical challenges of thermal technology that is of great concern [149,150]. Due to the nature of fast heat transmission, especially from the heating media, another important difficulty of pyrolysis to address is how to completely pyrolyze biowaste particles [5]. To produce high-quality products, the majority of pyrolysis reactors require small biomass particle sizes [149]. Insulation plays a crucial role in thermal energy conservation and assists the system in reaching even higher process temperatures. A product of higher quality can be produced at higher temperatures. These techniques aim to lessen emissions, specifically CO_2_ pollution. Nevertheless, there has not been a thorough investigation into the release of polluting gases. It will be advantageous to consider different chimney designs and configurations that could aid in the capturing of particulate carbon and lower environmental pollution in said technologies that needed improvement. The term “computational fluid dynamics” (CFD) refers to a class of computational methods for the studying of fluid and energy flows using numerical analyses [151]. The patterns of heat transfer in the reactor can be investigated using CFD. Understanding the main heat zones, the pattern of heat conduction and convection, and the potential for a synergistic heating effects on the carbonization chamber may be aided by this. Moreover, the biomass pyrolysis community also recognizes the issue of how to remove char fines from the liquid product as a concern [85]. There is still a need for consensus on how this can be performed easily at a low cost. In summary, design configuration, modeling, feedstock type, and the application of thermal conversion products have all been found to have knowledge gaps. It can be said that this method is an environmentally friendly technology for the thermal process-based products from biomass and that it may have significant effects on energy and environmental sustainability.

## 10. Concluding Remarks

Over the last decade, advances in the thermal conversion of potential feedstock, coupled with the application of suitable reactors for producing valuable products (particularly biochar, bio-oil, and pyrolytic gas), have garnered increased research interest. This is significant because thermochemical methods have provided viable pathways for converting low-value biowaste residues into important energy-based products, thus addressing globally significant energy. Biomass is increasingly seen as a potential source of alternative renewable energy. However, considering environmental concerns associated with their production, particularly regarding emission and waste control, recent years have witnessed advancements in thermochemical technologies for feedstock conversion to energy production. While several technologies and methods are still emerging, the attempt to lag/insulate the external body of the majority of the thermal technology needs further exploration. Technically speaking, thermal technology faces challenges related to the heat transfer from the source/particles of feedstock to fully pyrolyze, high product quality, and yield. Therefore, the high-pressure pyrolysis-based study is a novel solution to problems with insulation and product quantity/quality. The writers of this paper aimed to establish the fact that fluidized bed reactor types are more suitable and profitable among others, because those could enhance the product quality and quantity. Future research into high-pressure reactor designs and materials, along with promising feedstock varieties, is necessary to achieve further improvements in end product quality and quantity.

## Figures and Tables

**Figure 1 materials-17-00725-f001:**
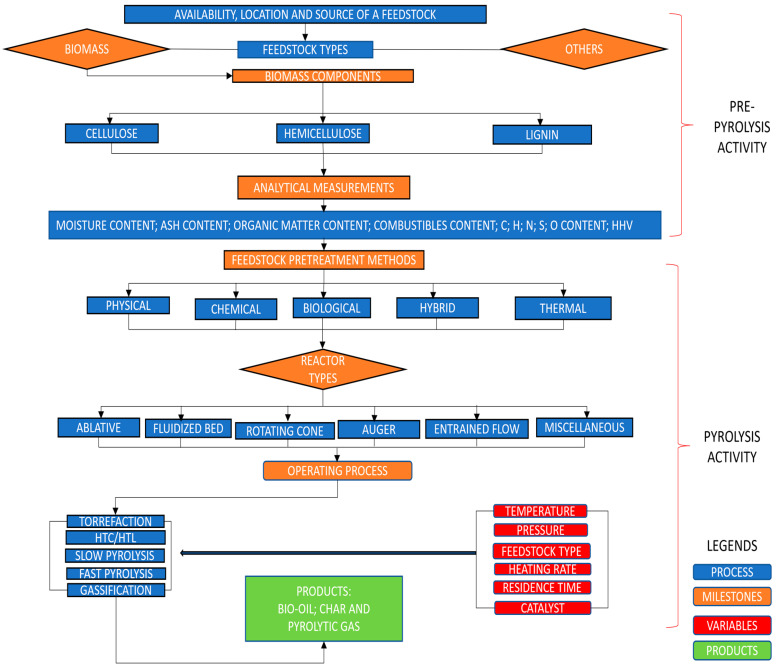
Key pre-to-main pyrolysis stages, from biomass selection, analytical measurements, treatment methods, to operating reactor-type processes output.

**Figure 2 materials-17-00725-f002:**
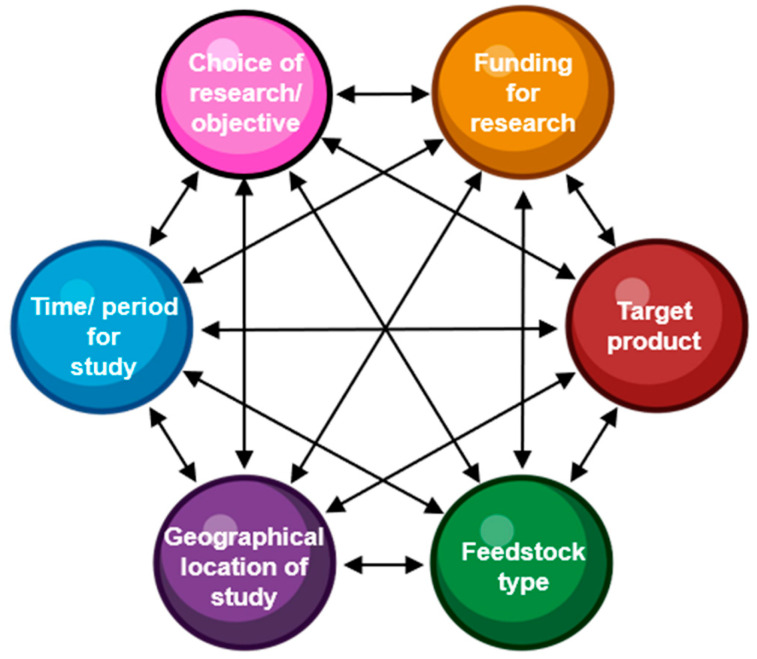
Potential inter-relatedness between individual components affecting pyrolysis-based research.

**Figure 3 materials-17-00725-f003:**
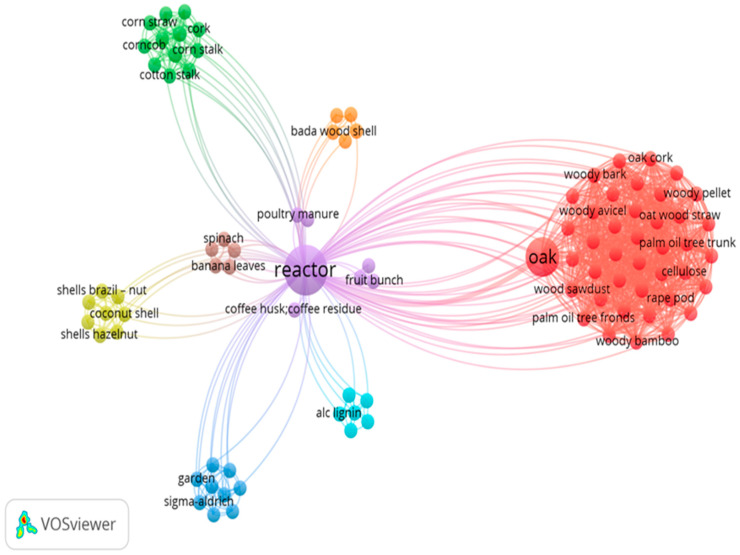
Occurrence mapping of different feedstock types (n = 97), which resulted in 10 groups.

**Figure 4 materials-17-00725-f004:**
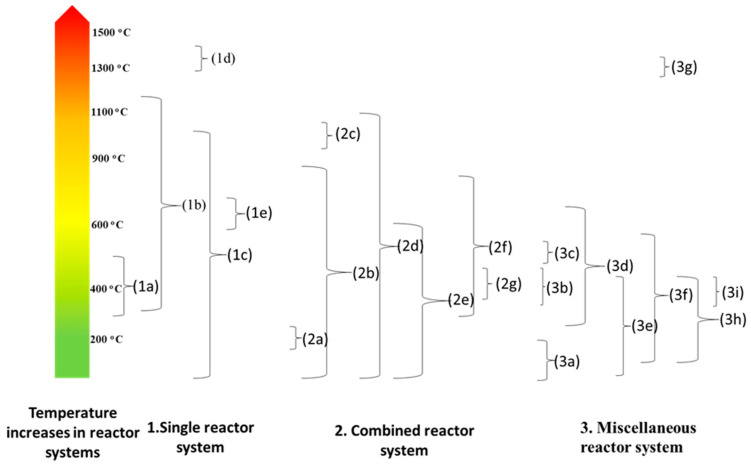
Schematic representation of pyrolysis temperature increases, reactors based on (1) single, (2) combined, and (3) miscellaneous operating systems. Here, the (1) single operating system includes: (1a) = fluidized bed reactor; (1b) = fixed-bed reactor; (1c) = TGA; (1d) = pressurized entrained flow gasification reactor; (1e) = ablative reactor. The (2) combined operating system includes: (2a) fixed bed with torrefaction; (2b) = TGA/FTIR; (2c) = TGA/DTG; (2d) = TGA/PyGCMS; (2e) = TGA/vertical dual bed tubular quartz; (2f) = analytical PyroProbe reactor; (2g) = TGA/DSC. (3) Miscellaneous reactors system include:|(3a) = hybrid organsolv steam; (3b) = Greenfield Eco. Pvt. Ltd. Pyrolyse instrument; (3c) = Cylindric furnace reactor; (3d) = HTC; (3e) = Muffle furnace; (3f) = Tubular quartz; (3g) = drop tube furnace; (3h) = semi-batch vertical; (3i) = continuous stirred tank reactor + plug flow reactor.

**Figure 5 materials-17-00725-f005:**
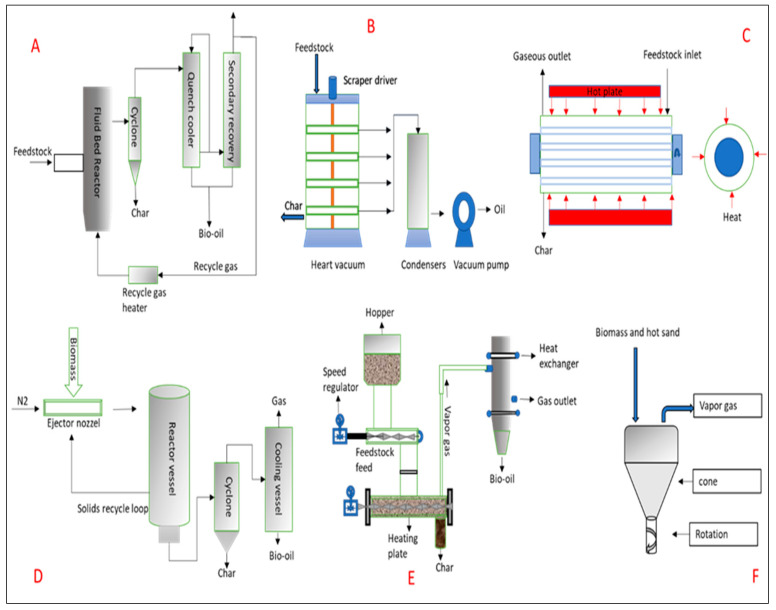
Thermochemical conversion technologies/reactors; schematic representation of thermochemical operating system based on reactor types. Where: (**A**) = Bubbling fluidized bed reactor; (**B**) = Vacuum reactor; (**C**) = Ablative reactor; (**D**) = Vortex reactor; (**E**) = Auger reactor; and (**F**) = Recirculating fluidized bed reactor. Adapted with modified from [5,127,137,138].

**Figure 6 materials-17-00725-f006:**
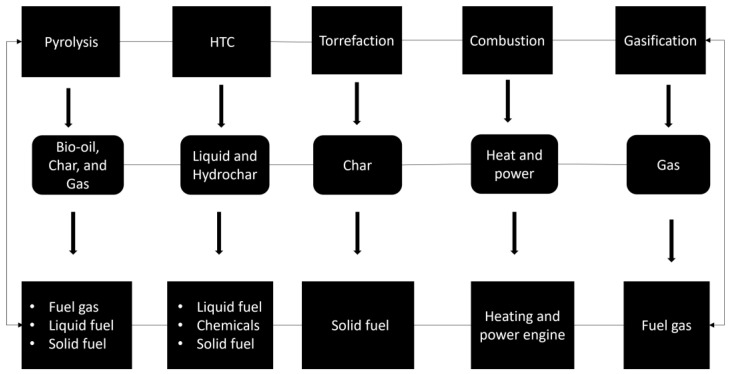
Thermochemical conversion technologies and expected products.

**Table 1 materials-17-00725-t001:** A summary of various experimental procedures of pyrolysis specific to their aims/objectives and analytical methods.

Aims/Objective	Methods	Ref.
Nine holocelluloses (two forestry and seven agricultural wastes) were selected as the feedstock to investigate the impact on the compositions of bio-oils and to screen the best feedstock suitable for the production of long-chain ethers precursor, for the ensuing improvement of yield and selectivity	Preparation of native holocellulose, evaluation of the sample, experimental apparatus, and procedures	[49]
To offer details on the yields and features of char produced from ten types of wood that are common in Southern Europe, undergoing biomass carbonization technologies condition	Biomass feedstocks, experimental facility, experimental procedure, charcoal characterization, and overview of the experiments	[50]
To perform intricate experimental analysis as well as the numerical modeling of oat straw’s slow pyrolysis. The pyrolysis products are described using advanced methods of analysis, with tests focusing on the properties and yield of the solid, liquid, and gaseous species	Feedstock sample, ultimate and proximate analyses and employing semi-batch vertical reactor where simultaneous thermal, infrared spectroscopy, qualitative of tars were analyzed, and pyrolysis gas analyzation, and numerical computations	[51]
The wet torrefaction of corn stalk was studied, and the biomass pyrolysis polygeneration performance of the wet torrefied sample was examined. More so, the solid material, energy, carbon, and hydrogen yields, as well as the effectiveness of removing ash and oxygen were also compared between WT and dry torrefied (DT) of corn stalks	Materials, torrefaction technique, characterization of torrefied samples, and pyrolysis technique	[52]
The determination of the thermal degradation characteristics of heating residues of eucalyptus (EU) and corncob (CC) for gasification using TGA rates of 10 °C/min in a nitrogen environment. The study covers the impact of biomass composition and kinetic parameters on heating rate	Preparation of biomass samples and experimental procedures	[32]
This experimental study set out to characterize the bioenergy potential of DS pyrolysis, measure gas emissions and byproducts, estimate kinetic and thermodynamic parameters, and detect the joint optimization of multiple responses in response to changing biofeedstock, heating rate, and temperature, as well as significant interactions between operational conditions	Sample preparation, physical and chemical analysis, TG experiments beforehand, activation energy, pyrolytic characteristic parameters, Friedman and Starink methods, Py-GC/MS experiments, TGA–FTIR experiments, and joint optimizations	[53]
To provide a thorough understanding of primary volatile compositions, mass loss behavior, reaction kinetics, and formation pathway during fast RH pyrolysis	Materials, pyrolysis process and kinetic methods	[31]
The impact of feedstock particle size on the distribution of fast pyrolysis products and the kinetics of slow pyrolysis	Characterization of MWSD, thermogravimetric analysis, evaluation of apparent activation energy, the pyrolysis of MWSD and product characterization, different profiles of mass loss and the impact of particle size on mass loss	[54]
To look into the reproducibility of TGA biomass pyrolysis experiments and potential deviations when mass loss kinetics are calculated from the same sample using various TGA technologies	TGA experiments and kinetic analysis	[33]
To fill the knowledge gap in orange and potatoes peel pyrolysis kinetics that was discovered during the literature review	Materials, TGA, and kinetics	[55]
To accurately evaluate the HHV using lumped-parameter pyrolysis kinetic models, and to demonstrate a straightforward correlation that can be used to assess HHV without relying on three different biomass species	Experimental samples, experimental procedures, and experimental results	[56]
Examine the combustion kinetics and study the combustion properties of five different types of biomass fuel pellets that can be used as biomass fuel	Analysis of the thermal weight loss and the components of five different biomass fuel pellet types	[44]
To investigate how the content of the biomass influences the kinetics, temporal evolution of the pyrolysis vapors, and production of the main bio-oil components during biomass pyrolysis	Materials, Py-FTIR analysis, isothermal mass loss of biomass, and using Py-GC/MS for the product analysis	[57]
To investigate the thermal decomposition of stalk and sour cherry flesh using thermogravimetric analysis, and to evaluate the activation energies using three kind of isoconversional approaches—Flynn–Wall–Ozawa, Friedman, or Kissinger–Akahira–Sunose. The findings reveal the pyrolysis kinetics and characteristics, as well as the ideal conditions for designing, optimizing, and simulating the pyrolysis process	Materials, physicochemical characterization, thermogravimetric analysis, and kinetic modeling	[58]
TGA/DTG investigation in an inert environment was performed to examine the thermal degrading and pyrolysis kinetics of biowastes.	Collection and preparation of biomass, proximate and ultimate investigation of samples as well as the calorific value, thermogravimetric/FTIR analysis	[59]
To carry out an extensive study that includes biochemical and physicochemical characterization, and the kinetic thermodynamic study of pyrolysis and thermal breakdown behavior of biomass from banana leaves	Sample preparation, banana leaves biomass pyrolysis reaction model determination using kinetic modeling, thermodynamic analysis, and thermogravimetric experiments	[60]
To clarify the pyrolytic behavior in terms of thermodynamic and kinetic characteristics, as well as the bioenergy potential of biological wastes resulting from the manufacturing of bio-products	The processing of bacterial biomass produced in a pilot-scale operation, sample characterization, FTIR spectroscopy, data processing using PCA, a TGA experiment, the characteristics of pyrolysis, thermo-kinetic studies pyrolysis, Py-GC/MS analysis, and the development of a model based on SVR	[61]
Pyrolyze three samples using thermogravimetric analysis and characterize them by determining how well various Phragmites Hirsuta components pyrolyze, thus this study offers theoretical direction for the formulation of the Phragmites preparation process, bioenergy is converted into Hirsuta by a thermochemical process	Material, characterization, Thermogravimetric analysis, kinetic modeling, reaction model determination, and thermodynamic analysis	[62]
To outline a straightforward method for analyzing the kinetic parameters (frequency factor, activation energy, and reaction model) of biomass with complicated thermal behavior. A multi-step mechanism for the biomass pyrolysis processes was employed to get the kinetic parameters using a deconvolution algorithm process coupled with isoconversional approaches.	Sample selection, preparation, and characterization, performed kinetics, and thermogravimetric analysis	[63]
In-depth research was conducted on the mechanisms causing the variations and the correlations between the pyrolysis characteristics and the various types of biomass. By improving our knowledge of the pyrolysis process in various biomass types, this work also serves as a reference for their thermal conversion methods	Materials, physicochemical of biomass, thermogravimetric, and kinetic analysis using the Coats–Redfern method TG and multi-peak fitting in the derivative thermogravimetric analysis.	[64]
Using a laboratory-scale (5 kg/h) AFP unit to accurately assess the impact of feedstock type on the characteristics of bio-oils produced from straw, miscanthus, and beech and poplar wood	Biomass that has been pyrolyzed, the pyrolysis process, the physicochemical characteristics of bio-oils, and a quantitative analysis of the chemical makeup of bio-oils	[6]
On the physical and chemical characteristics of biochar, particularly their effects on nitrogen (N) content and composition, the impact of feedstock type and temperature of pyrolysis were examined	Materials, preparation of biochar and sample preparation, and analytical methods	[65]
Studies involving feedstock, pyrolysis, and biochar, including policies on emission	Reviewing different concepts	[66]
The investigation of the effects of CaO addition sorbent and the temperature of pyrolysis on the chemical and the physical characteristics of obtained biochar and syngas	Material characteristics, experimental procedure, and methods	[38]
To look into how the structure of the resulting bio-char changed as the gaseous and liquid products evolved in relation to the pyrolysis temperature, and understanding how temperature affects the development of organics and the composition of biochar	Feedstock and chemicals, pyrolysis experiments, characterization of the products, and kinetic analysis	[43]
To ascertain how the duration time and pyrolysis temperature affect the properties of hydrochars in comparison to biochars produced through direct slow pyrolysis. In order to do this, hydrochar produced by HTC of waste biomass was pyrolyzed at two different temperatures (350 and 500 °C) and three different times (1, 3 and 5 h), and the testing was conducted to establish a number of properties relevant to the use of chars as soil amendment, inexpensive adsorbent, or fuel, and growing media, including pH, electrical conductivity, electrochemical potential, porosity, phytotoxicity, and elemental composition	Selection of hydrochar, pyrolysis of hydrochar made from waste biomass, pyrolysis of waste biomass, and char characterization	[67]
To investigate the impact of the pyrolysis temperature using fluidized bed pyrolysis system, three reactions were carried out to convert solid waste into renewable aviation fuel in attempt to show the distributions of the liquid and gas products at different temperatures	Feedstock, equipment, experimental procedures, and product analysis	[26]
The reaction mechanism of the co-pyrolysis of biomass and coal in the TGA analyzer was investigated using both conventional TGA and a novel congruent-mass TGA analyses. Studies that compare how these two approaches differ in how they assess the likelihood of a coal–biomass interaction	Materials and TGA	[47]
To research the kinetics of the co-pyrolysis of the coal and pretreated watermelon rind (WMR) blends	Selection of the biomass, pretreatment, compositional analysis, determination of the (WMR) higher heating value, calculation of its exergy, preparation of sample blends, thermogravimetric analysis of the coal and pretreated (WMR), kinetic analysis, and estimation of the thermodynamic parameters	[68]
The following research goals were achieved: (a) performing a thorough thermogravimetric analysis (TGA) of the nut shells; (b) identifying the characteristic points in the nut shells’ thermal decomposition process; (c) determining the temperature range at which hemicellulose, cellulose, and lignin decomposed in the examined nut shells; (d) estimating the fundamental kinetic parameters of the nut shells thermal decomposition; and (e) the physiochemical properties of the nut shells conversion rates as a function of the process temperature	Characteristics of the feedstock used in the research, thermogravimetric analysis, kinetic modelling, and model-fitting method: Coats–Redfern Method	[42]
TGA–FTIR (thermogravimetric analysis with FTIR analysis of evolved gases) pyrolysis experiment combined with advanced data analysis and modeling methods to assess the viability of developing an advanced methodology for the evaluation of biomass materials	Selection of the sample and testing on a suite of biomass materials	[41]
To assess the pyrolysis behavior of corks with various properties that might be used in scaling up the pyrolysis of cork-rich materials, in the strengthening of their value as well as their integration in thermochemical platforms	Materials, thermogravimetric analysis, kinetic analysis, estimation of chemical composition, wet chemical characterization, and FTIR analysis	[69]
The characteristics of green corn husks were described and analyzed in order to determine the thermokinetics conversion parameters through pyrolysis reactions that were kinetically studied using TGA and DTG, where the Flynn–Wall–Ozawa was used to compare the energetic efficiency from corn husk	Materials, biomass composition analysis, higher calorific value, non-isothermal thermogravimetric analyses, thermokinetics studies, master plots method, kinetic model proposed by Kissinger, kinetic model of Friedman, thermogravimetric analysis, and the mathematical simulation of the thermal decomposition kinetic of green corn husk biomass	[70]
To look into the technical and financial effects of different lignocellulosic elements on biomass pyrolysis, this work specifically investigates the basic mechanisms of cellulose, hemicellulose, and lignin transformation during pyrolysis	Characterization of biomass samples, sample preparation, pyrolysis, economic analysis, and validation via experimental values	[71]
To make available a theoretic framework for advancing the pyrolysis process and the efficient use of corn straw resources	Experimental materials, Instruments, and methods, analytical methods, and kinetics theory	[72]
Utilizing the pyrolysis poly-generation method to provide renewable energy and materials while overcoming the drawbacks of using rice husks	Materials, the preparation of an activated bio-char catalyst, a catalytic fast pyrolysis process, derived of amorphous SiO2 and porous carbon from bio-char, experiments on the adsorption of organic compounds, and physicochemical analysis	[34]
To research, ascertain, and comprehend these solids’ digestibility, as well as how the various hybrid method process parameters affected it	Feedstock and inoculum, pretreatment of wood chips, anaerobic digestion of pretreated solids and other analytical methods	[35]
In light of the fantastic outcomes produced in the chemical activation of rice husks (RHs), an assessment of bio-char made from RH pyrolysis was conducted to see if it could be used as a solid-phase extraction (SPE) to filter out harmful organic compounds from the biooil aqueous phase	Pyrolysis, chemical activation, characterization of activated carbon, SPE procedures, HPLC-DAD analysis, and method validation	[37]
Researchers have looked into the non-catalytic and catalytic co-pyrolysis of Ulva prolifera macroalgae (UPM) and straw (RS). To establish their ideal values, it has been investigated how temperature and mixing ratio affect the product’s distribution	Feedstock characterization, experimental setup and procedures, catalysts preparation, catalyst characterization methods, and liquid products analysis methods	[39]
Studies of techno-economic performance of involving biorefinery concepts and steam pretreatment techniques	Feedstock composition/economic analysis	[73]
Based on the composition of the ash, the investigation’s goal was to pinpoint the pertinent fractionation processes; the findings will later be applied to create a model for predicting slag composition and viscosity based on process parameters and fuel ash composition	Materials, feedstock preparation, and gasification process, and product char and gas analysis	[74]
To create the biofuel using a variety of techniques and examine the fuel’s characteristics	Pyrolysis, extraction of pyrolysis oil, gasification, and procedure for producer gas generation, the analysis of the coconut shell using TGA, ultimate analysis, producer gas composition, and proximate analysis	[45]
A comparative investigation on the two-step pyrolysis (TSP) of lignocellulosic biomass was carried out on samples of walnut shell (WS), cotton stalk (CS), corncob (CC), and their acid-washed counterparts using TGA–FTIR and Py-GC/MS	Materials and preparation, samples characterizations, and TGA–FTIR and Py-GC/MS analysis	[40]
To assess how relations between lignin and cellulose, which occur during the co-pyrolysis of lignin and cellulose at temperatures between 100 and 350 °C, affect char structure changes	Sample preparation, fast pyrolysis experiments, and sample characterization	[75]
To examine the viability of spent coffee grounds (SCG) upcycling via pyrolysis for the production of biochar and energy, while also proposing a circular economy scenario for the effective use of SGC produced in the city of Larisa, Greece	Materials characteristics, pyrolysis and process protocol	[48]
To determine the levoglucosan percentage in the bio-oils prepared from fast pyrolysis of hydrochloric acid-treated and untreated rice husks (RHs) under vacuum conditions	Materials, characterization of RHs, pretreatment of RHs, Fast pyrolysis procedure, bio-oil characterization, and quantification of levoglucosan in bio-oils	[36]
To research the impact of total pressure, pyrolysis temperature, and CO2 concentration on biomass char gasification at various temperatures	Biomass samples, char preparation, and char reaction models	[76]
To research (a) the influence of biochar made from mesquite on the combined physical and hydraulic properties of various compacted soils, and (b) the interdependence of hydraulic properties of biochar-amended soil on the physical properties for possible use in bioengineered structures	Biochar, soils, physical properties, hydraulic properties, FTIR, FESEM, XRD, BET, and statistical analysis	[77]
To investigate levoglucosenone (LGO) production used levoglucosan (LGA) as feedstock. LGA dehydration has a lower activation energy and is chemically simpler than cellulose pyrolysis, enabling the reaction to occur at low temperatures	Materials, reaction, and product analysis	[11]
To look into how pressure affects the pyrolysis of biomass’s thermal effects. Corn stalks, popular, switchgrass Trail-blazer, and switchgrass Alamo were the four energy crops chosen for experimental characterization	Materials, experimental techniques, and procedures	[78]
To assess the physicochemical potential of palm waste for pyrolysis processes that result in the production of biofuels	Preparation of biomass samples, and determination of physicochemical properties	[79]
To clarify differences and similarities among the combustion of the original raw biowaste and the combustion of bio-oil and biochar in order to better understand how fly ash forms during these processes	Biomass, biochar and bio-oil, fuel preparation prior to combustion experiments, combustion experiments, particle sampling system, operational procedure, and experimental plan, chemical analysis of the particulate matter, and multivariate data analysis are all covered in this study	[80]
To look into the possibility of preventing agglomeration and enhancing sugar formation during the pyrolysis of herbaceous biomass by combining ferrous, magnesium, and ammonium cations with sulfate anions	Methods for pretreatment, controlled pyrolysis duration-quench, continuous pyrolysis reactor system, assessment of sustainable throughput, quantification of sugar, ICP digestion, scanning, and electron microscopy analysis	[81]
To research the energy potential of hydrochar made from straw, Virginia mallow, and wood (pine) biomass. The hydrochars’ pyrolysis process was therefore investigated in order to determine how the gaseous byproducts changed with pyrolysis temperature	Materials, hydrothermal carbonization process, and pyrolysis	[82]
As an alternative technique for using waste biomass in the Polish context, a thorough study of slow solar pyrolysis of various waste biomass feedstock is presented. Although slow solar pyrolysis is the least expensive technology available due to the low heat input, it has the potential to produce highly porous solid fuels and provide a long-term solution for difficult waste disposal	Feedstock characterization includes determining the amount of lignocellulose in the feedstock as well as its ultimate and proximate analyses.sample preparation, sample analysis for C, H, and N, and BET surface area measurement of porosity	[83]
In order to comprehend pyrolysis behavior and potential interactions, investigations into the thermal decomposition of lignin and lignocellulosic biomass (watermelon rind) WMR were carried out at 325–625 °C to pyrolyze various lignin components in order to improve the pyrolytic products	Materials, experimental set-up and procedures, and product analysis	[84]

**Table 2 materials-17-00725-t002:** Constituents of some selected recent experimental works revealing pre-to-main pyrolysis stages, respectively, from biomass selection, and analytical methods, to biomass treatment method, reactor types, operating process, and product output.

Experimental Objectives	Pre-Pyrolysis			Main-Pyrolysis				Ref.
	Biomass Selection		Analytical Method	Biomass Treatment Method	Reactor Types	Operating Process	Product Output	
	One or More	Biomass Type	Moisture, Organic Matter, Ash Content, and Others					
To look into the yield and characteristics of the pyrolysis reaction products made from palm oil (trunk, frond, and shell) in an agitated reactor	Palm (trunk, frond and shell)	Palm tree	Moisture, ash content, and others	Physical and thermal	Agitated pyrolysis reactor, TGA, and DTA	Pyrolysis	Gas, bio-oil, and char	[103]
To investigate the influence of pyrolysis temperature (500–800 °C) on product yields in a conical spouted bed reactor with steam as a fluidizing source	Pine wood sawdust	Wood	Moisture, ash content, and others	Physical and thermal	Conical spouted bed	Pyrolysis	Gas, bio-oil, and char	[99]
Using steam pyrolysis of olive pomace, it was investigated how well various char-based catalysts (including biochar and coal char) produced hydrogen	Olive pomace	Olive	Moisture, ash content, and others	Physical and thermal	Fixed bed, TGA and others	Pyrolysis	Gas, bio-oil, char, and hydrogen	[100]
It was investigated how well the wet torrefied sample performed in the biomass pyrolysis polygeneration process as well as the WT of corn stalk	Corn stalk	Corn	Moisture, ash content, and others	Physical and thermal	Fixed bed	Pyrolysis	Biochar	[52]
Based on the characteristics of the pyrolysis process and its effectiveness in catalytic upgrading, the catalytic and non-catalytic pyrolysis of demineralized biowaste was examined and compared to raw biomass	Sawdust	Softwood	Moisture, ash content, and others	Chemical, physical and thermal	Fixed bed, Py-GC/MS,	Pyrolysis	Gas, bio-oil, and char	[102]
Examining the energetic, physical, and chemical characteristics of various biomass feedstocks in order to characterize their performances	Grapevine, olive trees, and others	Lignocellulosic residues	Moisture, ash content, and others	Physical and thermal	TGA	Pyrolysis	Bio-char and bio-fuel	[89]
To successfully scale up the pyrolysis process, it is crucial to thoroughly understand the effects of key variables on the devolatilization kinetics and bio-oil composition, such as biomass particle size, shape, content, heating rate, and residence period.	Saw dust	Wood	Moisture, ash content, and others	Physical and thermal	Pyroprobe^®^ 5200	Pyrolysis	Biochar and bio-oil	[54]
To ascertain the thermodynamic parameters and the kinetic triplet (activation energy, pre-exponential variable, and reaction model)	Banana leaves	Banana						[60]
Devoted to researching the online characterization, kinetic and thermodynamic analysis, thermal decomposition, and physicochemical characterization of hot vapors released during pyrolysis	Switchgrass	Crop	Moisture, ash content, and others	Physical and thermal	TGA-FTIR, Py-GC–MS examination	Pyrolysis	Gas, bio-oil, and char	[98]
This study looks at the effects of CaO on the evolution properties of cellulose, hemicellulose, and lignin pyrolysis products using TGA–FTIR and Py-GC/MS, and it also discusses the reaction mechanism of CaO-assisted pyrolysis of biowaste components	Cellulose and beechwood	Mixed	Moisture, ash content, and others	Chemical, physical and thermal	TGA–FTIR and PY-GC/MS	Pyrolysis	Bio-oil from	[25]
Using slow pyrolysis in a thermogravimetric analyzer, investigate the decomposition mechanism of the lab-scale grown microalga	Algal biomass	Algal	Moisture, ash content, and others	Physical and thermal	TGA	Pyrolysis	Biochar	[92]
(a) To methodically examine the recovery effectiveness of reducing sugars and VFAs at various HTS (4.17–8.28, 190–320 °C), and (b) to characterize the structure of the cornstalk following hydrothermal treatment at various HTS	Cornstalk	Corn	Moisture, ash content, and others	Physical	Batch	HTC	Volatile fatty acids (VFAs) and sugars	[88]
Determine the entrained flow reactor (EFR) used for the beech wood pyrolysis experiments, which were conducted at various gas residence times with temperature between 500 and 1400 °C. These experimental conditions were broad enough to produce chars with a range of characteristics	Beech	Wood	Moisture, ash content, and others	Physical and thermal	Entrained flow	Pyrolysis	Biochar	[15]
To investigate the characteristics of MD2 pineapple waste and its potential to become a feedstock for alternative solid biofuel	Pineapple	Pineapple	Moisture, fixed carbon content, and others	Physical and thermal	TGA	Pyrolysis	Biochar	[101]
To illustrate how canola residue may be a suitable biofuel feedstock for low-temperature (<450 °C) slow pyrolysis with energetically favorable conversions of up to 70 wt.% of volatile matter	Canola residue	Canola	Moisture, ash content, and others	Physical and thermal	TGA–FTIR	Slow pyrolysis	Bio-fuel	[105]
(1) To determine the transformation behavior of HMs during co-HTC, and (2) to investigate the fuel properties of the hydrochar from co-HTC. The results could provide support for SS utilization, particularly for fuel production with the targeted regulation of HMs	Sludge and biomass	Sludge and lignocellulosic	Moisture, ash content, and others	Physical	Autoclave reactor	HTC	Liquid and hydrochar	[90]
HTL thermal transformation of tobacco industry biowaste to oil in a multiple batch reactor	Tobacco	Tobacco	-	Physical	Batch reactor	HTL	Biocrude	[91]
Having in mind the literature presented on solar pyrolysis so far, a thorough study on slow solar pyrolysis of various waste biomass feedstocks is presented as an alternative method for using waste biomass in the Polish scenario, with a primary focus on fast and flash pyrolysis	Wood, stray sewage sludge	Mixed	Moisture, ash content, and others	Physical and thermal	Fixed-bed, TGA, and others	Pyrolysis	Gas, bio-oil, and char	[83]
To thoroughly investigate the catalytic potential of NZ (commonly found in Pakistan) in comparison to that of commercial ZSM-5 for raw and pretreated rice straw	Rice straw	Rice	Moisture, ash content, and others	Physical, chemical, and thermal	Fixed-bed	Pyrolysis	Gas and bio-oil	[93]
By combining acid impregnation and two-staged pyrolysis, the study aims to achieve staged and directional valorization of holocellulose and lignin in biomass waste	Eucalyptus waste	Wood	Moisture, ash content, and others	Physical, chemical, and thermal		Torrefaction and fast pyrolysis	Char, anhydrosugars, and phenols	[94]
In order to maximize utilization, it is important to compare specifically how well two common agricultural and forestry biomasses are suited for bioenergy production	Rice husk and poplar bark	Rice and wood	Moisture, ash content, and others	Physical and thermal	TG/DTG	Pyrolysis	Biochar	[95]
In particular, the effects on nitrogen (N) content and composition were examined, along with the impact of biomass type and pyrolysis temperature on the physical and chemical properties of biochar	Soybean straw and chlorella	Crop type	Moisture, ash content, and others	Physical and thermal	Stainless steel cylinder and electric muffle furnace	HTC/pyrolysis	Hydrochar and biochar	[65]
Study to lower energy consumption and increase glucose concentrations in enzymatic hydrolysis reactors	Wheat straw	Wheat	-	Chemical	Hydrolysis reactor	Hydrolysis and fermentation	Glucose	[104]
In the work, the catalytic activity of supported Al-containing bimetals was studied, and the synergy between the bimetals was discussed. In addition, the reaction pathways on the formation of furans were proposed	Corncob, wood, and others	Mixed	Moisture, ash content, and others	Physical and thermal	Py-GC × GC/MS	Pyrolysis	Furan	[96]
To research the microwave heating properties of coal gasification fine slag and its pyrolysis of biomass catalytic properties	Pine sawdust	Wood	Moisture, ash content, and others	Physical and thermal	Quartz tube, microwave-induced	Pyrolysis and gasification	Gas, bio-oil, and char	[97]

**Table 3 materials-17-00725-t003:** Summary of different biomass/feedstock as classified by various thermo-chemical reactors/analytical tools (considerations/factors of research objective).

Reactor/Analytical Tools		Biomass/Feedstock		Ref.
Mode	Types	Group Name	Group Examples	
Single	TGA	Woody	Eucalyptus	[32]
		Woody	Pellet	[44]
		Corn	Straw	[72]
		Walnut	Nut shell	[42]
		Hazelnut	Nut shell	[42]
		Pistachio	Nut shell	[42]
		Cork species	Cork	[69]
		Sugarcane	Bagasse	[63]
		Corn	Husk	[70]
		Wheat	Straw	[47]
		Woody	Bamboo	[47]
		Rice	Husk	[31]
		Woody	beech	[33]
		Peanut	Straw	[64]
		Sesame	Stalk	[64]
		Rape	Pod	[64]
		Tobacco	Stem	[64]
		Pecan	Shell	[64]
		Bada wood	Shell	[64]
		Woody	Camphor Tree	[64]
		Woody	Sapele	[64]
		Peanut	Straw	[64]
		Sesame	Stalk	[64]
		Woody	Poplar	[64]
		Woody	Willow	[64]
		Sour cherry	Stalk	[41]
		Sour cherry	Flesh	[41]
		*Phragmites hirsuta*	Root	[62]
		*Phragmites hirsuta*	Stem	[62]
		*Phragmites hirsuta*	Leaves	[62]
	Fixed bed	Rice	Husk	[34]
		Corn	Stalk	[34]
		Oak	Cork	[50]
		Oak	Holm	[50]
		Wood	Waste wood	[83]
		Herbaceous	Waste straw	[83]
		Sewage sludge	Sludge	[83]
		Woody	Anhydro sugar	[11]
		Model compounds	Cellulose	[43]
		Woody	Kraft	[75]
			Alkali	[75]
			Avicel	[75]
	Ablative	Woody	Poplar	[6]
			Straw	[6]
			Miscanthus	[6]
	Fluidized bed	Herbaceous	Corn stover	[81]
			Rice husk	[26]
	Entrained flow	Straw	Straw	[74]
	Furnace	Rice	Straw	[39]
		*Vival prolifera* macroalgae	*Vival prolifera* macroalgae	[39]
		Wood	Shavings	[66]
	Tubular quartz	Rice	Husk	[36]
	Adiabatic oxygen bomb calorimeter	Watermelon	Ring	[68]
	HTC	Parks	Park	[67]
		Gardens	Garden	[67]
	Wire mesh	Sigma-Aldrich	Sigma-Aldrich	[71]
	Semi-batch vertical	Oat wood	Straw	[51]
	DTG	Woody	Japanese cedar	[56]
		Woody	Castanopsis	[56]
		Rice	Straw	[56]
	Rotary-klin prototype	Plant	Coffee plant	[48]
	Furnace	Woody	Stem	[80]
		Woody	Bark	[80]
Combined	TGA and DSC	Corn	Stalk	[78]
		Switchgrass alamo	Grass	[78]
		Woody	Poplar	[78]
	TGA and STA	Banana	Leaves	[60]
	Pyro-Probe and CDS	Energy crop	Virginia mallow	[82]
		Woody	Pine	[82]
		Grass	Straw	[82]
	Gasifier and cylindrical reactor	Coconut	Shell	[45]
	TGA-FTIR and Py-Gc/MS	Corn	Cob	[40]
		Cotton	Stalk	[40]
		Walnut	Shell	[40]
	TGA and DTG	Orange	Peels	[55]
		Potato	Peels	[55]
		Coffee	Husk	[59]
		Coffee	Residue	[59]
		Palm oil tree	Fronds	[79]
			Shells	[79]
			Roots	[79]
			Trunk	[79]
	Fixed bed and quartz	Rice	Husk	[37]
	TGA, DTG and fixed bed	Leaves	Birch	[73]
		Wood	Spruce	[73]
	TGA and CDS	Wood	Sawdust	[54]
		Herbaceous	Corncob	[49]
			Wheat straw	[49]
			Rice husk	[49]
	Py-FTIR and Pyro-probe	Rice	Husk	[57]
		Woody	Pine	[57]
		Fruit	Bunch	[57]
	TGA and Py-GC/MS	Durian	Shells	[53]
	TGA-FTIR	Woody	Populus deltoides	[41]
			Pinus radiata	[41]
			Willow chips	[41]
			Roasted cashew nut	[41]
		Shells	Almond	[41]
			Hazelnut	[41]
			Brazil-nut	[41]
			Roasted cashew nut	[41]
	TGA–FTIR	Herbaceous	Reed canary grass	[41]
			Miscanthus giganteus	[41]
			Spinach	[41]
		Animal Product	Chicken manure	[41]
		Model compounds	cellulose (avicel)	[41]
			ALC lignin	[41]
			Xylan	[41]
			Dglucose	[41]
			Pectin	[41]
			Chlorogenic acid	[41]
	TGA and furnace	Herbaceous	Pine	[76]

**Table 4 materials-17-00725-t004:** Reactors with their properties.

Reactor Type	Technology Readiness	Advantages	Disadvantages	Ref.
Bubbling fluidized bed	Commercialized	Simplicity and ease to operation; efficient heat transfer; high bio-oil yield of 70–75%	Fine feedstock particles require	[127]
Circulating fluidized bed	Commercialized	suitable heat transfer, simpler scaling, and a useable particle size of 6 mm	More complex to operate and less liquid yield to achieve	[127,135]
Vacuum	Scaled up to about 3000 kg/h	No gas carrier is necessary, there are no complicated operating conditions, and it is possible to employ bigger biomass particles	Liquid yield (35–50%); large process equipment; slow heat transfer rate; greater coal content	[135]
Vortex	NREL	Particle sizes up to 20 mm, biomass particles were accelerated with high velocity, and yields of 65% liquids	High entering velocities of material into the reactor led to erosion at the transition from linear to angular momentum	[85]
Ablative	Laboratory scale	Larger particles may be used; there is no need for inert gas; heat transfer through hot reactor wall	Limitation on scale-up and heat supply issue	[138]
Auger	Pilot-scale, Understudy	Ceramic or still ball; sand as the heat carrier; mechanically driven	Bigger particles can be used; lesser liquid yield	[127]

## Data Availability

No new data were created or analyzed in this study. Data sharing is not applicable to this paper.

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
