# Peer review of "Navigating Pyrolysis Implementation—A Tutorial Review on Consideration Factors and Thermochemical Operating Methods for Biomass Conversion"

_materials, 2024, doi:10.3390/ma17030725_

Round 1

Reviewer 1 Report

Comments and Suggestions for Authors

1 The main content of this article is about the research on biomass pyrolysis. The introduction provided at the beginning is unrelated to the main content of the article and does not connect well with the following text.

2 The introduction should have mentioned the distribution, value, and current utilization status of biomass in relation to the selection methods. The selection methods for biomass would be more appropriate to be detailed in the subsequent content.

3 The content after line 111 in the introduction is suggested to be deleted.

4 The structure of the introduction is disorganized and the expression is unclear. The emphasis is inconsistent with the main focus of the article.

5 Table 1 in the second part, “The evolving nature of thermal conversion process,” should provide more detailed descriptions of the research methods, experimental conditions, and experimental conclusions.

6 The representation in Figure 2 is too complex and it is recommended to replace it

7 The experiments described in lines 220-230 do not align with the explanations provided.

8 Table 2 has the same issue as Table 1.

9 Section 7.8 can be merged.

Comments on the Quality of English Language

The author needs to polish the language well, and in addition, Many characters in the table have consistent capitalization

Author Response

The answers are included in the attached file.

Reviewer 2 Report

Comments and Suggestions for Authors

Comments to Białowiec et al.

Summary

The review inspects the principles of the pyrolysis process with the purpose of introducing beginners to the topic. The authors consider various aspects of the process from the planning to the implementation accounting for previous research into pretreatment methods, feedstock and reactor types.

General comments

The review is topical and its scope suitable for the Materials journal. Furthermore, the arrangement of the manuscript into sections and subsections is logical. As for the content, the manuscript is versatile and comprehensive. However, considering that the target group is beginners to the topic, the text could be more straightforward and structured. As it is now, it is quite difficult to navigate through a plethora of details. In addition, there is, at times excessive cautiousness in the formulation. If you are presenting new results based on your own research, it is wise to be modest in your statements, but in a review, where you are reviewing the current scientific insight, modesty should not overshadow clarity.

The tables play a central part in the manuscript, and they are sufficiently comprehensive. The figures are also of good quality, but they would need more support from the text.

Specific comments

2. The evolving nature of the conversion process: From the subsequent text, it is not very clear in what specific ways the conversion process is evolving in nature.

Figure 1: The YES and NO on both sides of the Feedstock types box near the top of the diagram seem superfluous. Furthermore, why are two of the items in ellipsoids and one in a rhomboid?

Figure 2: Almost a complete graph with edges between most of the node pairs. Why are there no arrows between funding for research and target products?

Line 188: The pyrolysis-based factors term seems a bit obscure. Of the factors mentioned on lines 186-188, the specific reactors item does not appear in figure 2 indicating it not being a pyrolysis-based factor, whereas the figure includes for example funding for research and feedstock types. What makes the two latter a pyrolysis-based factor contrary to the former?

Lines 203-204: What was proven?

Table 3: Is the last entry in the mode sub column empty?

Figure 3: The figure is certainly extravagant, but I am not sure it is very informative. For a start, what data does it illustrate? Furthermore, is it necessary to show all the connections to the reactor? It makes it difficult to spot any connections between the various clusters. Moreover, oak seems to be the only item, beside the reactor, with increased size in the occurrence mapping. Does it mean that oak is a particularly common feedstock for pyrolysis? What could be the reason for this special status. By the way, I am assuming that oak is not an acronym but refers to the wood species. Then, why are there no other woods in the mapping?

Reference 56: Unlike for the other references, the authors are in all capital letters.

Comments on the Quality of English Language

Not so much problem with the English language, but some of the formulations are unnecessarily florid.

Line 60: Furthermore.

Line 61: The use of the phrase more so in this way seems nonstandard. What is more, the production…

Line 70: The word and is here superfluous. (It is not connecting two items of the same grammatical type.)

Line 154: …along with how…

Table 1, page 6, second item from the top: The responses in response to formulation could be more concise.

Table 1, last entry on page 9: Either to look into or looked into.

3. Potential inter-relatedness of pyrolysis-based consideration factors: Rather hard to understand, bureaucratic language.

Line 240: I think the word resemble has a rather concrete meaning of being similar. Moreover, it is a transitive verb that needs an object. Instead, you could for example write that the feedstock and its pretreatment reflect each other or that they correspond, if the intention, as I assumed, is to say that one must modify the pretreatment according to the feedstock. If the intention was to say something else, please elaborate.

Line 247: There can be further studies, but to initiate a clause containing a new argument, the word furthermore is preferrable. 

Line 277: A summary of different…

Line 350: Inconsistency. From the left there is a reference to Boumanchar et al. as some workers, whereas from the right as a study

Line 908: Usually, it is better avoid having the body text referring to antecedents in the heading.

Author Response

(The authors gave the same response as above.)

Round 2

Reviewer 2 Report

Comments and Suggestions for Authors

The authors have responded and reacted to the questions of the reviewers making the necessary modifications. Hence, the manuscript is now easier to read and has improved sufficiently for publication.